# Formulation, optimization and sensitivity of NitrOMZv1.0, a biogeochemical model of the nitrogen cycle in oceanic oxygen minimum zones

Daniele Bianchi[1], Daniel McCoy[1], and Simon Yang[1]

[1]University of California Los Angeles, CA

**Correspondence:** Daniele Bianchi (dbianchi@atmos.ucla.edu)

**Abstract.**

Nitrogen (N) plays a central role in marine biogeochemistry by limiting biological productivity in the surface ocean, influencing the cycles of other nutrients, carbon, and oxygen, and controlling oceanic emissions of nitrous oxide ($N_2O$) to the atmosphere. Multiple chemical forms of N are linked together in a dynamic N cycle that is especially active in oxygen minimum zones (OMZs), where high organic matter remineralization and low oxygen concentrations fuel aerobic and anaerobic N transformations. Biogeochemical models used to understand the oceanic N cycle and project its change often employ simple parameterizations of the network of N transformations and omit key intermediary tracers such as nitrite ($NO_2^-$) and $N_2O$. Here we present a new model of the oceanic N cycle (**Nitr**ogen cycling in **O**xygen **M**inimum **Z**ones, or NitrOMZ) that resolves N transformation occurring within OMZs, and their sensitivity to environmental drivers. The model is designed to be easily coupled to current ocean biogeochemical models by representing the major forms of N as prognostic tracers, and parameterizing their transformations as a function of seawater chemistry and organic matter remineralization, with minimal interference with other elemental cycles. We describe the model rationale, formulation, and numerical implementation in a one-dimensional representation of the water column that reproduces typical OMZ conditions. We further detail the optimization of uncertain model parameters against observations from the Eastern Tropical South Pacific OMZ, and evaluate the model ability to reproduce observed profiles of N tracers and transformation rates in this region. We conclude by describing the model sensitivity to parameter choices and environmental factors, and discussing the model suitability for ocean biogeochemical studies.

# 1 Introduction

Nitrogen (N) limits phytoplankton production over large swathes of the ocean (Moore et al., 2013). Most of the N in the ocean is present as dissolved dinitrogen gas ($N_2$); however, only fixed N, e.g., ammonium ($NH_4^+$) and nitrate ($NO_3^-$), can be readily utilized by planktonic microorganisms, with the exception of N-fixing diazotrophs (Capone et al., 2008). The inventory and chemical form of N in the ocean are controlled by an active nitrogen cycle, whereby different chemical forms of the element are utilized as substrates for growth by a variety of microorganisms, either to supply building blocks for organic molecules, or to fuel metabolism via redox reactions (Capone et al., 2008; Kuypers et al., 2018). As a result, the residence time of fixed N in the ocean is on the order of 3,000 years or less, about one order of magnitude shorter than for the macronutrient phosphorous (Gruber and Galloway, 2008; Wang et al., 2019).

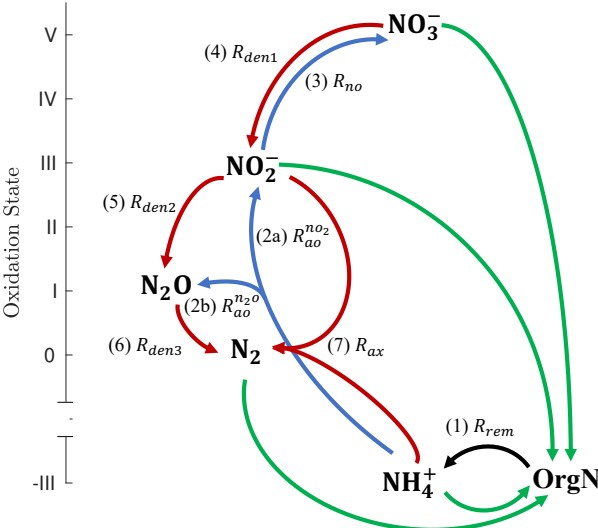

**Figure 1.** Schematic of the main N cycle tracers and reactions represented by NitrOMZ. Tracers are shown in bold, ordered by the oxidation state of N, and consist of organic nitrogen (OrgN), ammonium ($NH_4^+$), nitrate ($NO_3^-$), nitrite ($NO_2^-$), nitrous oxide ($N_2O$), and dinitrogen ($N_2$). N transformation reactions are shown as arrows connecting reactants and products. Green arrows represent nutrient uptake and nitrogen fixation rates (not explicitly represented in the model, which focuses on subsurface reactions). The black arrow corresponds to (1, $R_{rem}$) release of $NH_4^+$ by organic matter remineralization (i.e., ammonification). Blue arrows denote aerobic transformations: (2a, $R_{ao}^{no2}$) $NH_4^+$ oxidation to $NO_2^-$, (2b, $R_{ao}^{n2o}$) $NH_4^+$ oxidation to $N_2O$, and (3, $R_{no}$) $NO_2^-$ oxidation to $NO_3^-$. Red arrows represent anaerobic processes: (4, $R_{den1}$) $NO_3^-$ reduction to $NO_2^-$, (5, $R_{den2}$) $NO_2^-$ reduction to $N_2O$, (6, $R_{den3}$) $N_2O$ reduction to $N_2$, and (7, $R_{ax}$) anaerobic $NH_4^+$ oxidation (anammox) with $NO_2^-$ to $N_2$.

The ocean's inventory of fixed N is dominated by $NO_3^-$, and the main N cycle reactions consist of uptake and assimilatory reduction of $NO_3^-$ to $NH_4^+$ (here used interchangeably with ammonia, $NH_3$), and the oxidation of $NH_4^+$ back to $NO_3^-$ following the decomposition of organic matter and nitrification (Fig. 1). Only when the concentration of dissolved oxygen ($O_2$) drops to suboxic or anoxic levels (typically below 5 $\mathrm{mmol\ m^{-3}}$), additional metabolic pathways involving N become

relevant, as observed in the ocean's oxygen minimum zones (OMZs) and low-$O_2$ sediments (Lam and Kuypers, 2011). These reactions include the three main steps of heterotrophic denitrification, i.e., the oxidation of organic carbon (OrgC) with $NO_3^-$, nitrite ($NO_2^-$), and nitrous oxide ($N_2O$), and anammox, the chemolithotrophic oxidation of $NH_4^+$ with $NO_2^-$. Both denitrification and anammox lead to the production of $N_2$, and thus remove fixed N from the ocean (Bianchi et al., 2012; DeVries et al., 2012, 2013). Ammonia oxidation is another source of $N_2O$ — a powerful greenhouse gas and a leading agent of ozone

destruction in the stratosphere. The number of $N_2O$ molecules produced per $NH_3$ oxidized, i.e. the yield of this reaction, increases as $O_2$ declines (Goreau et al., 1980; Nevison et al., 2003) likely caused by a shift from $N_2O$ production as a byproduct of hydroxylamine oxidation to nitrifier-denitrification (Hooper and Terry, 1979; Wrage et al., 2001; Stein and Yung, 2003). Because of denitrification and enhanced production by ammonia oxidation, OMZs are important sources of $N_2O$ to the atmosphere (Naqvi et al., 2010; Yang et al., 2020), with the largest emissions observed right above shallow oxygen deficient waters

(Arévalo-Martínez et al., 2015).

The emerging picture for the ocean's N cycle is that of a web of inter-dependent transformations that is particularly active in OMZs, where overlapping aerobic and anaerobic reactions exchange nitrogen metabolites and substrates (Lam and Kuypers, 2011; Kuypers et al., 2018), ultimately controlling fixed nitrogen removal and $N_2O$ production. While there is evidence that organic matter and $O_2$ regulate the rates and relative importance of N transformations (Babbin et al., 2014; Dalsgaard et al.,

2014), our mechanistic understanding of these environmental controls against the backdrop of oceanic variability remains limited. Ocean biogeochemical models can shed light on the expression of the N cycle reactions in a dynamic environment. These models have included N as a macronutrient since the beginning, representing $NO_3^-$ and $NH_4^+$ assimilation by phytoplankton and subsequent nitrification (Fasham et al., 1990; Sarmiento et al., 1993; Moore et al., 2004b). With the advent of more complex Earth System Models, biogeochemical representations have progressively expanded to include more detailed

representations of the N cycle, including N fixation, denitrification and $N_2O$ production (Aumont et al., 2015; Séférian et al., 2020; Stock et al., 2020; Long et al., 2021b).

The ultimate goals of these models are multifold, and include improving predictability of oceanic $N_2O$ emissions (Suntharalingam et al., 2012; Martinez-Rey et al., 2015; Battaglia and Joos, 2018; Buitenhuis et al., 2018; Ji et al., 2018a); providing a more realistic representation of the redox state of seawater (Louca et al., 2016); or resolving aspects of microbial dynamics

underlying the oceanic N cycle (Penn et al., 2016; Zakem et al., 2018; Penn et al., 2019).

The representation of N transformations in models often relies on crude assumptions that simplify the network of N reactions and their controls to simple empirical parameterizations. For example, models that include $N_2O$ cycling often rely on parameterizations that link $N_2O$ production to nitrification or aerobic respiration (Suntharalingam and Sarmiento, 2000; Nevison et al., 2003; Manizza et al., 2012; Jin and Gruber, 2003), overlooking $N_2O$ sources and sinks by denitrification. These models

also conflate anammox and denitrification into a single $N_2$ production term. Explicit cycling of $NO_2^-$ under low $O_2$, with the observed co-occurrence of $NO_2^-$ production from $NO_3^-$ dissimilatory reactions, reduction to $N_2O$ and $N_2$ by denitrification and anammox, and reoxidation to $NO_3^-$ are missing (Lam and Kuypers, 2011; Kalvelage et al., 2013; Babbin et al., 2014, 2015; Buchwald et al., 2015a; Babbin et al., 2017).

The goal of this paper is to present a new model of the oceanic N cycle designed to be incorporated in current ocean biogeo-chemical models, with a particular focus on processes occurring within OMZs. We refer to this model as NitrOMZ (**Nitr**ogen cycling in **O**xygen **M**inimum **Z**ones). The model explicitly represents the major forms of N found in seawater as prognostic tracers, and parameterizes the transformations that connect them as a function of seawater chemistry. This formulation is informed by recent observations that describe the response of N cycle reactions to environmental controls, in particular the availability of substrates and dissolved $O_2$. We detail the implementation of the model in an idealized one-dimensional (1D) representation of the water column that allows comparison to in situ observations, formal optimization, and studies of the model sensitivity to parameter choices and environmental conditions.

The rest of the paper is organized as follow: Sect. 2 discusses the rationale and formulation of the model; Sect. 3 the implementation of the model; Sect. 4 the model optimization against tracer and rate observations; Sect. 5 the performance of the model and its sensitivity to environmental parameters; and Sect. 6 discusses the implications and conclusions of the work.

## 2   Nitrogen cycle model formulation

### 2.1   Model rationale

The NitrOMZ model is based on the current understanding of the N cycle in OMZs (Lam and Kuypers, 2011; Kuypers et al., 2018) as mediated by 6 major species: $N_2$, $NO_3^-$, $NO_2^-$, $N_2O$, $NH_4^+$ and organic nitrogen (OrgN) in either dissolved or particulate form. We only explicitly model $NH_4^+$ (the dominant dissolved form) and do not distinguish it from $NH_3$. We also assume that organic nitrogen is linked to organic carbon by fixed stoichiometry (Anderson and Sarmiento, 1994), although variable stoichiometry can easily be accommodated.

A schematic of the model's tracers and transformation is shown in Fig. 1. Our approach represents a natural progression for current biogeochemical ocean models, and takes a "system view" of the N cycle by focusing on the biogeochemistry of N transformation reactions (Lam and Kuypers, 2011), rather than microbial ecology (Penn et al., 2016; Louca et al., 2016; Zakem et al., 2018; Penn et al., 2019). That is, the model explicitly resolves N chemical tracers and their transformations, but not the populations of microbes that are responsible for these reactions.

The underlying assumption is that the occurrence and rates of N transformations are controlled by, and can be predicted from, the physical and chemical conditions of the oceanic environment. Implicitly, the model assumes that diverse populations of microbes are always present in the water column, and that their activity (i.e., metabolic rate) is controlled by the abundance of substrates, in analogy to chemical reactions, and dissolved $O_2$, which inhibits anaerobic reactions (Kalvelage et al., 2011; Babbin et al., 2014; Dalsgaard et al., 2014; Ji et al., 2018a; Sun et al., 2021b). The focus on dissolved N forms and reaction rates bypasses poorly-known aspects of microbial population dynamics, which are topics of ongoing research (Louca et al., 2016; Zakem et al., 2018; Penn et al., 2019).

We assume that each reaction is implicitly mediated by specialized microorganism groups, each relying on a distinct metabolism (Lam and Kuypers, 2011; Kuypers et al., 2018). Thus, the model represents a "modular" N cycle, with individual reaction steps (i.e., individual redox reactions) represented separately, and connected by exchange of dissolved substrates (Graf et al., 2014; Kuypers et al., 2018). This premise is grounded on observations of high specialization and streamlined genomes for marine prokaryotes (Giovannoni et al., 2014), including microorganisms carrying genes for N-based metabolic reactions (Ganesh et al., 2015; Kuypers et al., 2018).

These assumptions are sufficient to provide a broad representation of microbial N transformations and their environmental expressions in the ocean, while limiting model complexity and the proliferation of poorly-constrained parameters. They are also grounding steps towards models that explicitly represent microbial populations, including their diversity and dynamics in OMZs (Louca et al., 2016; Penn et al., 2016; Zakem et al., 2018; Penn et al., 2019).

### 2.2   Model tracers and processes

The model focuses on microbial processes that take place below the euphotic zone, as driven by the flux of organic matter produced near the surface and exported into the ocean interior by the biological pump (Boyd et al., 2019). We include heterotrophic and chemolithotrophic pathways that are commonly observed in the open ocean and require N species as substrates

(Kuypers et al., 2018) (Fig. 1). Additional pathways, for example involving sulfur or iron, could also be represented following a similar approach.

Heterotrophic reactions resolved by the model (Fig. 1) consist of aerobic organic matter respiration ($R_{rem}$, pathway 1), which relies on $O_2$ as the oxidant, and the three main steps of denitrification: dissimilatory $NO_3^-$ reduction to $NO_2^-$ ($R_{den1}$, pathway 4), $NO_2^-$ reduction to $N_2O$ ($R_{den2}$, pathway 5), and $N_2O$ reduction to $N_2$ ($R_{den3}$, pathway 6). Chemolithotrophic processes consist of aerobic oxidation of $NH_4^+$ to both $NO_2^-$ ($R_{ao}^{no2}$, pathway 2a) and $N_2O$ ($R_{ao}^{n2o}$, pathway 2b via both hydroxylamine oxidation and nitrifier-denitrification), aerobic oxidation of $NO_2^-$ to $NO_3^-$ ($R_{no}$, pathway 3), and anammox, the anaerobic oxidation of $NH_4^+$ with $NO_2^-$ to produce $N_2$ gas ($R_{ax}$, pathway 7). Reactions are parameterized as functions of substrates (i.e., model tracer concentrations) and environmental parameters such as dissolved $O_2$ and organic matter. Tracers are expressed as concentrations, with units of $\mathrm{mmol\,m^{-3}}$.

We do not include an explicit representation of nitric oxide, NO, because of the poor understanding of its cycle in the marine environment (Ward and Zafiriou, 1988). NO is thought to be an obligate intermediate or a byproduct of N cycle reactions, including nitrification and denitrification (Schreiber et al., 2012). However, it is a very reactive chemical with extremely low concentrations (on the order of $\mathrm{pmol\,m^{-3}}$) and rapid turnover in seawater (Ward and Zafiriou, 1988). As a consequence, in situ NO observations are limited (Lutterbeck et al., 2018), and rate measurements targeting NO reactions are missing. Impicitly, we assume that NO cycles so rapidly that accumulation and transport by the oceanic circulation are negligible, and that its dynamics can be folded into the cycle of other N tracers.

There are also several notable processes that are not represented in the current model formulation, but could be introduced in future releases. Some of these processes (e.g., dissimilatory $NO_2^-$ reduction to $NH_4^+$, DNRA) are not thought to be quantitatively relevant in oceanic oxygen minimum zones. Others, while relevant, require further measurements to constrain their significance and response to environmental variability.

Production of $N_2O$ via $NH_4^+$ oxidation in NitrOMZ is represented as a single $O_2$-dependent function designed to model the transition in bacterial metabolisms from predominantly hydroxylamine oxidation to nitrifier-denitrification at low $O_2$ (Hooper and Terry, 1979; Wrage et al., 2001; Stein and Yung, 2003; Nevison et al., 2003). However, growing evidence suggests that ammonia oxidizing archaea (AOA, which greatly outnumber their bacterial counterparts) can also produce $N_2O$ via a hybrid mechanism (Santoro et al., 2011; Löscher et al., 2012). Production of $N_2O$ via AOA appears to be similarly enhanced at low $O_2$ (Trimmer et al., 2016; Santoro et al., 2021), although evidence from Stieglmeier et al. (2014) argues otherwise.

DNRA, which can be dominant in anoxic sediment, has been sporadically observed in the water column of oxygen deficient zones, where it may provide an additional source of $NH_4^+$ to anammox bacteria (Lam et al., 2009; Lam and Kuypers, 2011; Kraft et al., 2011; Jensen et al., 2011). However, DNRA is commonly undetectable in OMZ waters (Kalvelage et al., 2013; De Brabandere et al., 2014) , and its importance to the N cycle of OMZ is still debated (Long et al., 2021a).

Recent tracer incubation studies show substantial and often dominant formation of $N_2O$ from $NO_3^-$ rather than $NO_2^-$ (Ji et al., 2018b; Frey et al., 2020). This suggests that denitrifying bacteria capable of direct production of $N_2O$ from $NO_3^-$ reduction (as $NO_2^-$ reduction proceeds entirely within the cell) could be a major source of $N_2O$. This idea, which contrasts with the model assumption of a fully "modular"' N cycle, is further supported by isotopic evidence (Casciotti et al., 2018).

Observations needed to constrain the proportion of $N_2O$ from $NO_3^-$ and $NO_2^-$ and its environmental sensitivity remain however limited (Ji et al., 2018b; Frey et al., 2020).

Other work suggests the occurrence of $NO_2^-$ oxidation in apparently $O_2$-deficient waters (Buchwald et al., 2015b; Babbin et al., 2020; Sun et al., 2021a). This may involve $NO_2^-$ oxidation coupled to iodate reduction, or $NO_2^-$ disproportionation — two poorly characterized processes. It may also reflect the high affinity to $O_2$ of nitrite-oxidizing bacteria (Bristow et al., 2016) in regions where vanishing $O_2$ concentrations are maintained by infrequent lateral intrusions (Buchanan et al., 2023).

Finally, the model could easily accommodate missing processes that couple the N cycle with other elemental cycles, in 150 particular carbon and sulfur. These include formation of organic matter by chemolithotrophy, changes in inorganic carbon chemistry (e.g., pH) by anaerobic reactions (Cinay et al., 2022) , and additional metabolic pathways such as anaerobic oxidation of sulfide with $NO_3^-$ (Callbeck et al., 2021), and anaerobic oxidation of methane with $NO_2^-$ (Thamdrup et al., 2019), both chemolithotrophic denitrification reactions.

## 2.3   Model equations

Heterotrophic reactions (i.e., organic matter remineralization) are parameterized as a function of the respective oxidants and organic matter concentration, and expressed in carbon units per unit volume and time. Heterotropic reaction rates are assumed to be first-order in the concentration of organic matter, and limited by the oxidant following a Michaelis-Menten formulation (Johnson and Goody, 2011). Anaerobic reactions are inhibited by the presence of $O_2$, based on an exponential limitation term (Dalsgaard et al., 2014). The resulting equation for a general heterotropic reaction is:

$$R_H = k_H \cdot \frac{[X]}{[X] + K_H^X} \cdot e^{-\frac{O_2}{K_H^{o2}}} \cdot POC \tag{1}$$

Here, $H$ indicates the heterotrophic process considered (e.g., dissimilatory reduction of $NO_3^-$ to $NO_2^-$), $R_H$ the heterotrophic reaction rate (mmol C m$^{-3}$ s$^{-1}$); $k_H$ the specific first-order reaction rate (s$^{-1}$); $[X]$ the concentration of the oxidant (i.e., $O_2$, $NO_3^-$, $NO_2^-$ or $N_2O$); $K_H^X$ the half saturation constant for oxidant uptake (mmol m$^{-3}$); $K_H^{o2}$ the scale for inhibition of the reaction by $O_2$ (mmol m$^{-3}$); and POC the concentration of particulate organic matter in units of mmol C m$^{-3}$. No $O_2$ 165 inhibition is applied to aerobic respiration (i.e., $K_H^{o2}$ can be thought of as arbitrarily large).

Chemolithotropic reactions are proportional to the respective substrates. A maximum reaction rate is modulated by the concentration of oxidants and reductants, following Michelis-Menten dynamics. For anaerobic reactions (here, anammox), an $O_2$-dependent inhibition term limits the reactions when $O_2$ is present. The resulting equation for a general chemolithotrophic reaction is:

$$R_A = k_A \cdot \frac{[X]}{[X] + K_A^X} \cdot \frac{[Y]}{[Y] + K_A^Y} \cdot e^{-\frac{O_2}{K_A^{o2}}}. \tag{2}$$

Here, $A$ indicates the chemolithotropic process considered (e.g. anammox), $R_A$ the reaction rate (mmol N m$^{-3}$ s$^{-1}$); $k_A$ the maximum reaction rate when the process is not limited (mmol N m$^{-3}$ s$^{-1}$); $[X]$ and $[Y]$ the concentrations of the oxidant and reductant respectively (e.g., $NO_2^-$ and $NH_4^+$ for anammox); $K_A^X$ and $K_A^Y$ the half saturation constants for oxidant and reductant

uptake respectively ($mmol\ m^{-3}$); and $K_A^{o_2}$ the scale for inhibition of the reaction by $O_2$ ($mmol\ m^{-3}$). For aerobic reactions, $K_A^{o_2}$ is set to infinite, removing $O_2$ inhibition.

Equations for each of the heterotrophic and chemolithotrophic reactions are presented in Appendix A1 and A2, respectively; parameter names, units, and suggested values from the literature are presented in Table 1.

## 2.4 Model assumptions and parameterizations

In the model, we assume that heterotropic reactions are first-order to the concentration of organic matter, thus all organic matter can be utilized by microorganisms without saturation at high concentrations. Because of the low abundance of organic matter in seawater, and extensive colonization of particles by heterotrophic bacteria, this is a reasonable first-order assumption. However, see Nguyen et al. (2022) for a discussion of microbial-particle interactions in ocean biogeochemical models, and more complex aspect of their dynamics. For simplicity, we represent organic carbon by a single component. This assumption is easily relaxed to include multiple carbon species, for example separate particulate or dissolved forms.

We do not explicitly model conversion of dissolved $CO_2$ to organic matter by chemolithotrophy, because of the small rates compared to the remineralization of organic matter in the upper ocean. This assumption can also be relaxed in future implementations of the model, allowing a more complete integration between chemolithotrophy and the carbon cycle.

The use of an exponential inhibition term for anaerobic reactions by $O_2$ is based on the observation that they are limited at $O_2$ concentrations of few $mmol\ m^{-3}$ or smaller (Dalsgaard et al., 2014; Babbin et al., 2015; Frey et al., 2020). However, coexistence of anaerobic and aerobic reactions at $O_2$ concentrations of 10–20 $mmol\ m^{-3}$ or higher is also observed (Kalvelage et al., 2011), perhaps related to the presence of redox microenvironments within organic particles (Bianchi et al., 2018; Smriga et al., 2021), which are not explictly considered here. The exponential inhibition formulation has the advantage of being controlled by a single parameter, allows anaerobic reactions at finite $O_2$ concentrations, and approximates empirical rates from incubation experiments reasonably well (Dalsgaard et al., 2014).

Parameter values for maximum reaction rates, half saturation constants, and $O_2$ inhibition terms (Eq. (1) and (2)), are informed by analysis of previous work, and further optimized against in situ observations of tracers and rates (Sect. 4). Table 1 presents a list of the model parameters and measured values based on a review of the literature. Note that these studies are based on shipboard and laboratory incubations that differ in the setup, conditions, and microbial populations tested. Despite these caveats, experimental results provide valuable starting points to further constrain parameter values in the model.

**Table 1.** Summary of the main NitrOMZ parameters, with any reported values from the literature (not available or 'N/A' otherwise).

| Parameter | Description | Units | Value(s) | Source(s) |
|---|---|---|---|---|
| $k_{rem}$ | Maximum aerobic remineralization rate | $d^{-1}$ | 0.08 | Babbin et al. (2015) |
| $k_{ao}$ | Maximum $NH_4^+$ oxidation rate | $\mu mol\ N\ m^{-3}\ d^{-1}$ | $45.56 \pm 4.8$ | Bristow et al. (2016) |
|  |  |  | $37.10 \pm 1.2$ | Peng et al. (2016) |
|  |  |  | 20 - 100 | Ward (2008) |
| $k_{no}$ | Maximum $NO_2^-$ oxidation rate | $\mu mol\ N\ m^{-3}\ d^{-1}$ | $255.5 \pm 17.3$ | Bristow et al. (2016) |
|  |  |  | $63.3 \pm 13.8$ | Sun et al. (2017) |
|  |  |  | $55.7 \pm 5.4$ | Sun et al. (2017) |
| $k_{den1}$ | Maximum $NO_3^-$ reduction rate | $\mu mol\ N\ m^{-3}\ d^{-1}$ | $197 \pm 26$ | Bristow et al. (2016) |
| $k_{den2}$ | Maximum $NO_2^-$ reduction rate | $\mu mol\ N\ m^{-3}\ d^{-1}$ | N/A | N/A |
| $k_{den3}$ | Maximum $N_2O$ reduction rate | $\mu mol\ N\ m^{-3}\ d^{-1}$ | N/A | N/A |
| $k_{ax}$ | Maximum anammox rate | $\mu mol\ N\ m^{-3}\ d^{-1}$ | N/A | N/A |
| $K_{rem}^{o2}$ | $O_2$ half saturation constant for oxic remineralization | $mmol\ O_2\ m^{-3}$ | 4 | Babbin et al. (2015) |
| $K_{ao}^{nh4}$ | $NH_4^+$ half saturation constant for $NH_4^+$ oxidation | $mmol\ N\ m^{-3}$ | $0.134 \pm 0.005$ | Martens-Habbena et al. (2009) |
| $K_{ao}^{o2}$ | $O_2$ half saturation constant for $NH_4^+$ oxidation | $mmol\ O_2\ m^{-3}$ | $0.33 \pm 0.13$ | Bristow et al. (2016) |
|  |  |  | $3.6 \pm 0.6$ | Peng et al. (2016) |
| $K_{no}^{no2}$ | $NO_2^-$ half saturation constant for $NO_2^-$ oxidation | $mmol\ N\ m^{-3}$ | $0.254 \pm 0.161$ | Sun et al. (2017) |
| $K_{no}^{o2}$ | $O_2$ half saturation constant for $NO_2^-$ oxidation | $mmol\ O_2\ m^{-3}$ | $0.778 \pm 0.168$ | Bristow et al. (2016) |
| $K_{den1}^{no3}$ | $NO_3^-$ half saturation constant for $NO_3^-$ reduction | $mmol\ N\ m^{-3}$ | N/A | N/A |
| $K_{den2}^{no2}$ | $NO_2^-$ half saturation constant for $NO_2^-$ reduction | $mmol\ N\ m^{-3}$ | N/A | N/A |
| $K_{den3}^{n2o}$ | $N_2O$ half saturation constant for $N_2O$ reduction | $mmol\ N\ m^{-3}$ | N/A | N/A |
| $K_{ax}^{nh4}$ | $NH_4^+$ half saturation constant for anammox | $mmol\ N\ m^{-3}$ | N/A | N/A |
| $K_{ax}^{no2}$ | $NO_2^-$ half saturation constant for anammox | $mmol\ N\ m^{-3}$ | N/A | N/A |
| $K_{den1}^{o2}$ | $O_2$ exponential inhibition for $NO_3^-$ reduction | $mmol\ O_2\ m^{-3}$ | $1.05 \pm 0.72$ | Bristow et al. (2016) |
| $K_{den2}^{o2}$ | $O_2$ exponential inhibition for $NO_2^-$ reduction | $mmol\ O_2\ m^{-3}$ | $0.429 \pm 0.2$ | Dalsgaard et al. (2014) |
|  |  |  | $2.16 \pm 1.3$ | Ji et al. (2018a) |
| $K_{den3}^{o2}$ | $O_2$ exponential inhibition for $N_2O$ reduction | $mmol\ O_2\ m^{-3}$ | $0.27 \pm 0.05$ | Dalsgaard et al. (2014) |
| $K_{ax}^{o2}$ | $O_2$ exponential inhibition for anammox | $mmol\ O_2\ m^{-3}$ | $1.28 \pm 0.6$ | Dalsgaard et al. (2014) |
| $Ji_a$ | Nevison et al. (2003) 'a' parameter | % | $0.11 \pm 0.05$ | Santoro et al. (2021) |
|  |  |  | $0.2 \pm 0.13$ | Ji et al. (2018a) |
| $Ji_b$ | Nevison et al. (2003) 'b' parameter | % | $0.077 \pm 0.07$ | Santoro et al. (2021) |
|  |  |  | $0.08 \pm 0.04$ | Ji et al. (2018a) |

## 3 Model implementation

### 3.1 One-dimensional model setup

We implement the model for a 1D water column that includes physical transport by vertical advection and turbulent diffusion (Wyrtki, 1962), and, if required, parameterized lateral transport by horizontal currents and eddies (Gnanadesikan et al., 2013; Bettencourt et al., 2015). The model is configured to represent the typical weak upwelling conditions that characterize open-ocean oxygen minimum zones, following previous work (Babbin et al., 2015).

In the 1D framework, the conservation equation for the concentration [C] of a generic dissolved tracer can be written as:

$$\frac{\partial [\text{C}]}{\partial t} = -\frac{\partial (w_u \cdot [\text{C}])}{\partial z} + \frac{\partial}{\partial z} K_v \frac{(\partial [\text{C}])}{\partial z} + \sum_{i=1}^{N_H} \left( r_{C,H}^i \cdot R_H^i \right) + \sum_{i=1}^{N_A} \left( r_{C,A}^i \cdot R_A^i \right) + LT \tag{3}$$

Here, $w_u$ is the vertical upwelling velocity (m s$^{-1}$) and $K_v$ is the vertical turbulent diffusion coefficient (m$^2$ s$^{-1}$, distinct from molecular diffusion which is much smaller), both of which can be a function of depth. The first and second summation are respectively over the $N_H$ heterotrophic and $N_A$ chemolithotrophic processes that involve the tracer (Eq. (1) and (2)), with $r_{C,H}^i$ and $r_{C,A}^i$ the corresponding stoichiometric ratios (Appendix A4). $LT$ represents any parameterized lateral transport process. The explicit equations for each of the model tracers are detailed in Appendix A5.

The lateral transport terms $LT$ can be included to parameterize horizontal circulation by advection and diffusion in the 1D framework. Typically, these terms are simplified by a linear restoring to far-field tracer concentration profiles (Babbin et al., 2015), $[\text{C}]^{far}$, with a relaxation timescale $\tau_C$ (s):

$$LT = -\frac{1}{\tau_C} \cdot \left( [\text{C}] - [\text{C}]^{far} \right) \tag{4}$$

For typical open ocean conditions, $\tau_C$ can be estimated as the minimum of an advective timescale $\frac{L}{U}$ and a diffusive timescale, $\frac{L^2}{K_H}$, where $L$, $U$, $K_H$ are respectively the horizontal spatial scale, the horizontal velocity scale, and the horizontal eddy diffusion. Assuming $L$ on the order of 1000 km, $U$ on the order of 0.01 m s$^{-1}$, and $K_H$ on the order of 1000 m$^2$ s$^{-1}$ results in a timescale $\tau_C = 10^8$ s, i.e., on the order of 3 years and in agreement with recent estimates of the residence time of water within the Eastern Tropical South Pacific (ETSP) (Ji et al., 2015b; Johnston et al., 2014).

### 3.2 Organic matter remineralization

In the 1D model implementation, we represent organic matter (OrgC and OrgN) as a single particulate organic carbon (POC) class that sinks through the water column. We assume that this sinking is rapid compared to advection and diffusion, leading to a steady-state distribution of POC that is only controlled by sinking and remineralization (Kriest and Oschlies, 2008). Since remineralization rates are proportional to the concentration of organic matter, the resulting steady-state 1D equation for POC is:

$$\frac{\partial (w_s \cdot \text{POC})}{\partial z} = -\sum_{i=1}^{N_H} R_H^i = \sum_{i=1}^{N_H} \left( k_H^{eff,i} \right) \cdot \text{POC} \tag{5}$$

Here, $w_s$ is the depth-dependent sinking speed of POC in the water column, and $k_H^{eff,i}$ ($s^{-1}$) are the effective rate constants for each heterotrophic process, i.e., the maximum rate constants multiplied by the respective substrate limitation and $O_2$ inhibition terms (Eq. (1)).

Considering the flux of sinking POC, $\Phi_{POC}$ ($mmol\,C\,m^{-2}\,s^{-1}$):

$$\Phi_{POC} = w_s \cdot POC, \tag{6}$$

equation 5 can be written as:

$$\frac{\partial \Phi_{POC}}{\partial z} = -\sum_{i=1}^{N_H} R_H^i = -\sum_{i=1}^{N_H} \left( k_H^{eff,i} \right) \cdot POC, \tag{7}$$

or, equivalently,

$$\frac{\partial \Phi_{POC}}{\partial z} = -\frac{\sum_{i=1}^{N_H} \left( k_H^{eff,i} \right)}{w_s} \cdot \Phi_{POC} \tag{8}$$

Equation 7 can be recast to relate the concentration of POC in the water column to the remineralization of the POC flux with depth:

$$POC = -\frac{1}{\sum_{i=1}^{N_H} \left( k_H^{eff,i} \right)} \cdot \frac{\partial \Phi_{POC}}{\partial z} \tag{9}$$

The advantage of Equation 9 is that it allows to diagnose sinking POC concentrations when the POC flux and remineralization rate constants are known. In the 1D implementation of the model, we parameterize the POC flux following a typical depth-dependent power-law function, or Martin curve (Martin et al., 1987; Berelson, 2001; Primeau, 2006):

$$\Phi_{POC} = \Phi_{POC}(z_0) \cdot \left( \frac{z}{z_0} \right)^{-b}, \tag{10}$$

where $z_0$ is the upper boundary of the model, and $b$ the power-law or Martin coefficient. A plot of the model POC is shown in Fig. C1. Another advantage of this formulation is that it allows coupling NitrOMZ to more complex parameterizations for the remineralization of organic matter in ocean biogeochemical models, some of which rely on explicit representation of sinking organic particles, and some of which only represent sinking organic particle fluxes in the water column (Moore et al., 2004a; Dunne et al., 2010; Aumont et al., 2015; Stock et al., 2020; Long et al., 2021b). Because NitrOMZ's equation can be cast as a function of prescribed vertical organic matter flux or remineralization profiles, the model can be coupled to existing biogeochemical models with minimal interference with their formulation of organic matter cycles.

### 3.3 Numerical implementation of the 1D model

For the purpose of testing and illustration, we implement NitrOMZ in a 1D representation of the water column below the mixed layer, following previous work (Babbin et al., 2015). Model tracers are discretized on a 1D vertical grid, with equal spacing $\Delta z = 10\,m$, where $z$ is depth. Boundary conditions are set at the top ($z_0$) and bottom grid ($z_{bot}$) cells, as Dirichlet (or

fixed concentration) boundary conditions, with values taken from observations (Tables B2 - B3). The conservation equation for each tracer (following Eq. (3), see Appendix A5 for full equations) is then solved using a forward in time, centered in space numerical scheme, with a constant vertical grid spacing, and the option for a variable or constant timestep. In the baseline simulations (Fig. 2), we adopt a timestep of 5 days for the initial 650 year spinup, and decrease it to 3 hours for the final two years of the simulation (years 698 and 699) to increase accuracy.

As in Babbin et al. (2015), NitrOMZ does not represent primary production in the surface layer, and is instead forced at the uppermost boundary by a flux of sinking POC, $\Phi_{POC}(z_0) = w_s(z_0) \cdot POC(z_0)$, where $POC(z_0)$ provides the boundary condition for POC. The flux $\Phi_{POC}$ remineralizes in the water column based on a Martin curve profile (Eq. (10)). At each depth, the steady-state conservation equation for POC (Eq. (8)) is solved with a forward-in-space method, using a depth-dependent sinking speed $w_s$ chosen to produce, together with the maximum aerobic remineralization rate constant, $k_{rem}$, a POC flux profile matching a Martin curve with exponent $b$ appropriate for the oxygenated ocean (Primeau, 2006; Weber and Bianchi, 2020). To this end, the sinking speed is calculated at each depth as:

$$w_s = \frac{k_{Rem} \cdot z}{b} \tag{11}$$

The concentration of POC in the water column is then diagnosed using Equation 9, and used to calculate the heterotrophic remineralization rates $R_H$ in Equation 1 (see Appendix A1).

Under constant forcings and boundary conditions, the model tracers evolve towards steady-state ($\frac{\partial [C]}{\partial t} \approx 0$, Fig. 2) with a timescale $\tau_{SS}$ that can be estimated from the advection velocity $w_u$, the turbulent vertical diffusion $K_v$, and the vertical scale $H$, as the minimum between $\frac{H}{w_u}$ and $\frac{H^2}{K_v}$. For $w_u$ on the order of 10 m y$^{-1}$, $K_v$ on the order of $10^{-5}$ m$^2$ s$^{-1}$, and a vertical scale of 1000 m, the timescale to approach steady-state is $\tau_{SS} = 3 \cdot 10^{10}$ s, or about 100 years.

Fig. 2 shows an example of model spinup to steady state in NitrOMZ, with parameters taken from an optimal solution discussed in Sect. 5.2, and uniform initial tracer concentrations in the water column. At the start of the simulation, high water column O$_2$ leads the aerobic remineralization ($R_{rem}$) to dominate total POC consumption. As the simulation proceeds, an O$_2$ minimum develops in subsurface waters, reaching suboxic ($< 10$ mmol O$_2$) concentrations around year 100. NO$_3^-$ reduction rates ($R_{den1}$) are relieved of O$_2$ inhibition and begin to take up a larger fraction of total POC remineralization, as revealed by the depletion of N$^*$, signaling NO$_3^-$ consumption in the water column. Reduction of NO$_3^-$ also leads to a subsurface peak in NO$_2^-$ within the O$_2$-minimum (Fig. 2). With newly available NO$_2^-$ substrate and low O$_2$ conditions, NO$_2^-$ reduction ($R_{den2}$) begins, resulting first in a subsurface spike in N$_2$O. With further decrease in O$_2$ concentrations, N$_2$O is reduced to N$_2$, leading to a layer of low N$_2$O concentrations within the OMZ that persists to the end of the simulation. Anammox ($R_{ax}$) is similarly relieved of O$_2$ inhibition as the O$_2$-minimum is established, reaching maximum values near the upper oxycline, reflecting relatively high supply of both NO$_2^-$ and NH$_4^+$.

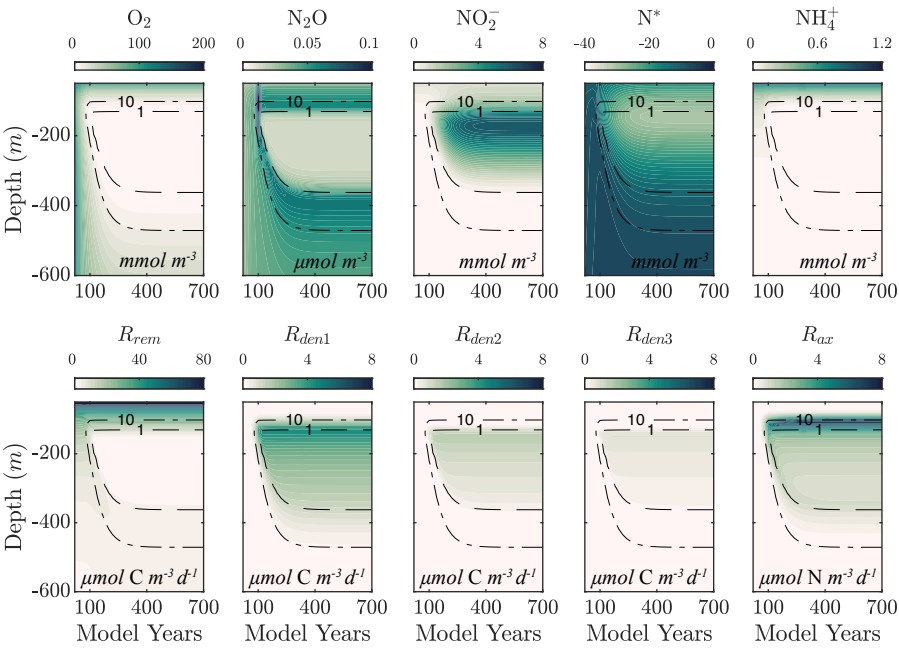

**Figure 2.** Example of spinup of the model. (top) Temporal evolution of $O_2$, $N_2O$, $NO_2^-$, $N^*$, and $NH_4^+$ from initial ETSP boundary conditions at year 0 to the final model solution at year 700 using the selected parameter set ($Opt_{sel}$) discussed in Sect. 5.2. Dashed black curves highlight the 1 and 10 mmol $O_2$ m$^{-3}$ contours. (bottom) Same as (top), but for the heterotrophic rates of aerobic respiration ($R_{rem}$), $NO_3^-$ reduction ($R_{den1}$), $NO_2^-$ reduction ($R_{den2}$), and $N_2O$ reduction ($R_{den3}$). The chemolithotrophic anammox rate ($R_{ax}$) is also shown in the far right panel.

## 4  Model optimization strategy

The model contains 23 major parameters that control the N cycle, some of which are relatively well constrained by observations, whereas others are poorly known and can plausibly span a broad range of values (Table 1). In the model, these parameters approximate complex or poorly known aspects of microbial physiology, metabolism and ecology, and thus are intrinsically
uncertain. In order to select a set of parameters that produces a realistic representation of the N cycle in OMZ, we adopt a "metaheuristic" approach based on application of an optimization algorithm, following an established strategy in ocean biogeochemistry (Schartau and Oschlies, 2003; Ward et al., 2010; Kriest et al., 2017).

To conduct this optimization, we compile available tracer and biogeochemical rate observations for the ETSP OMZ from a July 2013 cruise aboard the R/V Nathaniel B. Palmer, for which abundant trace and rate measurements are available (Fig.
5) (Ji et al., 2015b; Peng et al., 2016; Babbin et al., 2017, 2020), as well as from other cruises in the region (Kalvelage et al., 2013). The observations are then used to define a cost function based on normalized squared deviations between model profiles and observations. The cost function is minimized by applying a Covariance Matrix Adaptation Evolutionary Strategy algorithm (CMA-ES, discussed in Section 4.1), which finds a local optimal solution in the model's multi-dimensional parameter landscape.
The optimization is characterized by large dimensionality, strong non-linearity, a significant computational cost (requiring several 10,000s model runs to converge), and inherent flexibility in the formulation of the cost function (Schartau and Oschlies, 2003; Kriest et al., 2017). Thus, instead of seeking a single global optimal solution, we generate an ensemble of optimal solutions that provide equally acceptable representations of OMZ processes based on the cost function. To this end, we apply the optimization multiple times, varying the formulation of the cost function slightly, and assigning a random error to the
observations for each optimization (Table B4). As a result, we produce a set of equally plausible optimal solutions that we further evaluate to select a final parameter set based on additional comparisons with observations, which we use for further analysis.

### 4.1  Optimization algorithm

The CMA-ES is a stochastic, population-based algorithm that seeks to minimize an objective cost function (Hansen et al.,
2009). The CMA-ES falls within the broader class of evolutionary optimization algorithms, where search for an optimal solution proceeds by an iterative improvement of a population of parameters, with each iteration including a stochastic "evolutionary" element, in loose analogy with biological processes of mutation, recombination, and selection (illustrated in Fig. 3). In contrast with typical evolutionary computation algorithms such as Genetic Algorithms, in the CMA-ES the mutation and recombination operations are substituted by sampling from a multivariate normal distribution in which parameters (the
covariance matrix) are deterministically updated based on previous iteration steps (Hansen, 2006).

The CMA-ES has been shown to be more efficient (i.e., requiring fewer objective function evaluations), accurate (i.e., able to approximate the global optimum when it is known to exist), and robust (i.e., not overly sensitive to the initial choice of parameters), compared to other optimization algorithms, when applied to multi-dimensional, non-linear optimization problems

(Hansen et al., 2009; Hansen, 2023). These properties make it suitable for optimization of ocean biogeochemical models (Kriest
et al., 2017). A detailed description of the algorithm procedure can be found in Hansen (2023); an overview of the main steps
of the algorithm, and its application to ocean biogeochmsitry are presented in Kriest et al. (2017).

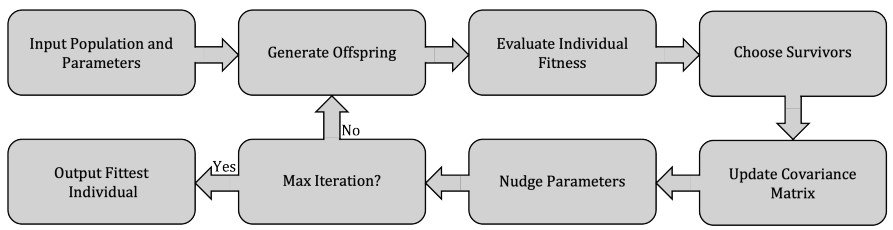

**Figure 3.** Flowchart of the CMA-ES optimization algorithm used to constrain uncertain model parameters.

## 4.2 Optimization implementation

As an illustration of NitrOMZ, we perform a series of optimizations against ETSP OMZ observations. For this configuration,
we set a constant upwelling velocity ($w_{up}$) but impose a variable vertical diffusion ($K_v$) profile, with lower diffusion in upper
stratified layers, and a transition to higher diffusion in deeper layers (Fischer et al., 2013) (Fig. C1, left panel). This is a
simplifying assumption that allows us to control the vertical scale for advective-diffusive transport (given by the ratio between
vertical diffusivity and upwelling velocity, $\frac{K_v}{w_{up}}$), without requiring vertical divergence terms in the conservation equation for
tracers associated with variable vertical velocities. Since this simulation targets the core of the OMZ, generally characterized
by sluggish horizontal circulation (Karstensen et al., 2008), we turn off far-field tracer restoring. This simplifies analysis of
model balances between transport and reaction rates, while resulting in realistic tracer distributions. Top and bottom boundary
conditions are listed in Table B3 and are extracted from observations.

As a first step, we select parameters that control aerobic remineralization processes ($R_{rem}$) and lead to a realistic vertical $O_2$
profile relative to ETSP observations, including the vertical position and thickness of oxygen deficient waters ($O_2 < 5$ mmol
m$^{-3}$) (Fig. 5). These consist of the vertical diffusion and upwelling magnitude, the Martin curve coefficient ($b$), and the upper
ocean POC flux ($\Phi_{poc}^{top}$), based on values consistent with observations (Table B2 and Fig. C1). For simplicity, we also set the
maximum aerobic remineralization rate ($k_{rem}$) and the $O_2$ half saturation constants for $NH_4^+$ and $NO_2^-$ oxidation ($K_{ao}^{o_2}$ and
$K_{no}^{o_2}$, respectively) to reported values in the literature (see Table 1). We then employ the CMA-ES algorithm in NitrOMZ to
optimize the remaining 20 parameters that control heterotrophic and chemolithotrophic reactions in Fig. 1, using the range of
parameter values listed in Table B1.

To optimize more uncertain parameters that control the anaerobic N cycle, we then conduct 4 sets of optimizations, with
cost functions devised to match desired characteristics of tracer and rate profiles in the ETSP OMZ. Briefly, the cost function
is calculated as the mean square of the difference between observations and model output profiles for a series of variables
that include tracers and N transformation rates (listed in Table B4). Before each optimization, a random error of up to 20%

is assigned to each observation to increase the variability of observational constraints and improve the robustness of the op-

timization ensemble by preventing from always converging in the neighborhood of a specific local minimum controlled by non-relevant features of the observations. Three additional constraints are imposed to improve the fit to observations for N cycle processes occurring within the core of the OMZ. First, all rates are weighted equally, whereas different weights are assigned to each tracer, giving higher weight to $N_2O$ and $NO_2^-$, which are central to the anaerobic N cycle. Because of possible influence from horizontal advection in observations, discrepencies exist between modelled and observed $NO_3^-$ and $PO_4^{3-}$. To

compensate for this, we also assign lower weights to $NO_3^-$ and $PO_4^{3-}$, and higher weight to $N^*$. Second, a depth-dependent weighting scheme is included to emphasize the match to observations in the OMZ interior. This vertical weight is shaped as a Gaussian curve centered at the core of the observed OMZ, where the bulk of anaerobic transformations targeted by our model occurs, so that values within the core of the OMZ are weighted up to twice as much as values outside the OMZ. Finally, N cycle transformation rates are shifted vertically to match their depth relative to the oxycline (here defined as $O_2$=1 mmol m$^{-3}$)

in both model and observations, and rescaled by a factor proportional to observed vs. modeled POC flux in the upper ocean. The only difference between the 4 sets of optimization are the relative weights assigned to each tracer, listed in Table B4. In total, we obtain 382 optimized parameter sets for further analysis.

## 5 Results and sensitivity

### 5.1 Optimization results

The distributions of the parameter values from the 382 sets of optimizations (see Sect. 4.2 and Table B4) are shown in Fig. C2. Rather than converging always to the same set of parameters, the optimization shows some variability for specific parameters. This reflects the stochastic nature of the CMA-ES algorithm, the inclusion of random variations in the observations, and the highly non-linear nature of the optimization problem, which may allow for non-unique optimal solutions. Optimized maximum rates (such as $k_{ao}$, $k_{no}$, $k_{den1}$, and $k_{den3}$) and exponential $O_2$ inhibition parameters for step-wise denitrification ($K^{o2}_{den2}$ and

$K^{o2}_{den3}$) reveal more variability than half-saturation concentration coefficients ($K$ terms), which often settle to the minimum or maximum allowed value (Table B1).

Pairwise correlations in Fig. 4 reveal several parameters pairs which exhibit strong relationships, reflecting the fact that, in a significantly non-linear optimization, similar results can be obtained by trade-offs between different parameters and processes. Notably, the exponential $O_2$ inhibition constants for $NO_2^-$ and $N_2O$ reduction ($K^{o2}_{den2}$ and $K^{o2}_{den3}$, respectively) are strongly

correlated with each other (R = 0.73) and with other parameters controlling the denitrification steps. These include positive correlations with the maximum rate parameters for $NO_3^-$ and $NO_2^-$ reduction ($k_{den1}$ and $k_{den2}$, respectively) and negative correlations with the half-saturation constants for $NO_2^-$ and $N_2O$ reduction ($K^{no2}_{den2}$ and $K^{n2o}_{den3}$, respectively). These correlations suggests tight couplings between modelled denitrification steps, wherein high/low maximum denitrification rates can be compensated by lower/higher half-saturation coefficients, respectively.

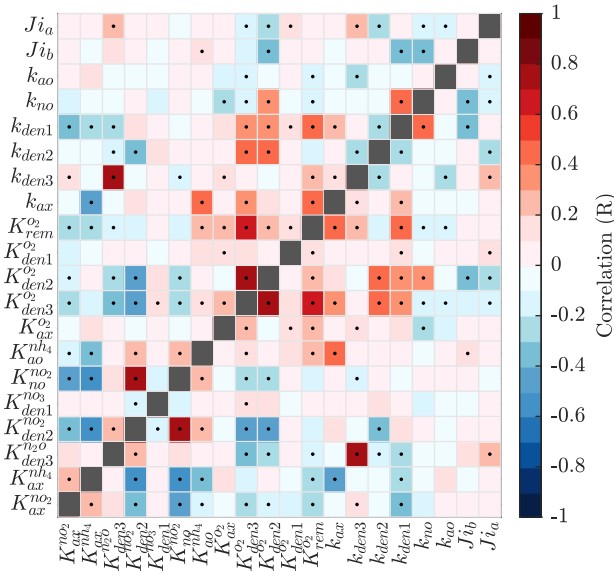

**Figure 4.** Pairwise correlations between model parameters for model solutions optimized for the ETSP OMZ. See Table B1 for a list and a description of the model parameters. Correlation is shown as the Pearson correlation coefficient, with dots representing $p$-values $< 0.01$.

Considering the variability in the optimal parameter sets, and the complexity of the cost function, which depends on observations for multiple variables at different depths, the resulting N-cycle profiles show similar features across all optimal solutions (Fig. 5, top panels; see also Fig. C3 for macronutrient profiles). When compared to observations, the majority of parameter sets are able to skillfully model (1) the vertical distribution of $O_2$, including the oxygen deficient layer between roughly 100 to 400 m, (2) the subsurface maximum in $NO_2^-$, (3) the rapid attenuation of $NH_4^+$ with depth, and (4) the subsurface minimum in $N^*$.

N cycle transformation rates also show similar consistency in their vertical profiles, albeit with more notable discrepancies with observations, possibly reflecting the higher variability and more complex nature of these measurements. Lower rates than observed may also reflect the fact that incubation experiments provide potential rates, rather than in situ rates. In general, the yield of $N_2O$ from $NH_4^+$ oxidation ($R_{ao}^{n2o}$) is $\mathcal{O}(100)$ times less than for $NO_2^-$ ($R_{ao}^{no2}$) following Equations A8 and A9, consistent with observations (Ji et al., 2015a, 2018a; Santoro et al., 2021). The step-wise denitrification rates ($R_{den1}$, $R_{den2}$, and

$R_{den3}$) show remarkably similar vertical profiles, with higher $NO_3^-$ reduction rates ($R_{den1}$) and nearly identical magnitudes between $R_{den2}$ and $R_{den3}$. Anammox ($R_{ax}$) shows a similar profile as denitrification, albeit with enhanced local maxima near the upper and lower oxycline depths surrounding the OMZ core, consistent with observations (Kalvelage et al., 2013).

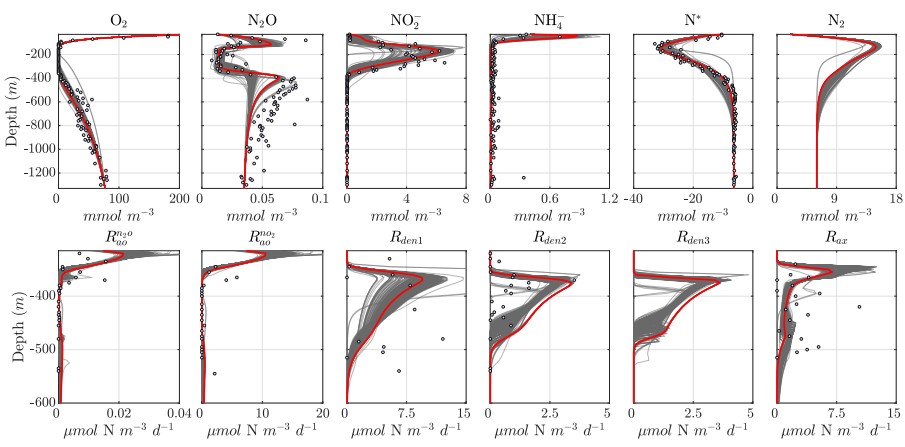

**Figure 5.** Results from the optimized ensemble of model solutions. (top) Tracer ($O_2$, $N_2O$, $NO_2^-$, $NH_4^+$, $N^*$, and $N_2$) profiles from all 382 optimized ETSP parameter sets. The bold red curves show the selected parameter set ($Opt_{sel}$) discussed in Sect. 5.2. Observations used to define the optimization cost function are shown as circles in each panel. Macronutrient profiles ($NO_3^-$ and $PO_4^{3-}$) are shown in Fig. C3. (bottom) Same as in (top), but for reaction rate profiles of $N_2O$ and $NO_2^-$ production from $NH_4^+$ oxidation ($R_{ao}^{n2o}$ and $R_{ao}^{no2}$, respectively), $NO_3^-$ $NO_2^-$, and $N_2O$ reduction ($R_{den1}$, $R_{den2}$, and $R_{den3}$, respectively) and anammox ($R_{ax}$).

     Several robust features emerge from the optimized parameter solutions, suggesting underlying mechanisms that need to be captured for a faithful representation of the OMZ N cycle. In particular, the differences in the exponential $O_2$ inhibition

parameters for denitrification, shown in Fig. 6 (left panel), reveal the existence of progressively lower $O_2$ tolerance for step-wise denitrification ($K_{den3}^{o2} < K_{den2}^{o2} < K_{den1}^{o2}$) from all optimized parameter sets. As a result, denitrification can stop at either $N_2O$ or $NO_2^-$ as $O_2$ increases above anoxic levels, leading to "incomplete" denitrification (Babbin et al., 2015).

Within the anoxic core of the OMZ ($\sim$100 to 350 m depth), $O_2$ is low enough in all optimizations to allow each of the steps to proceed unimpeded (Fig. 5). The large differences between $NO_3^-$ and $NO_2^-$ reduction ($R_{den1}$ - $R_{den2}$, middle panel of Fig. 6) allows accumulation of a characteristic subsurface peak in $NO_2^-$ near the OMZ core. Conversely, $N_2O$ produced via $NO_2^-$ reduction ($R_{den2}$) is quickly consumed via $N_2O$ reduction ($R_{den3}$) leading to a pronounced $N_2O$ deficit near the OMZ core. The progressive $O_2$ inhibition of the three steps of denitrification results in a decoupling between these reactions that is particularly evident in the oxycline layers above and below the OMZ, where $N_2O$ accumulation dominates as $N_2O$ reduction (i.e., consumption) is more strongly inhibited by $O_2$ than $NO_2^-$ reduction (i.e., $N_2O$ production, right panel of Fig. 6). Thus, the $O_2$ range defined by $K_{den2}^{o2}$ and $K_{den3}^{o2}$ can be thought of as a $N_2O$-production "window" that allows net $N_2O$ accumulation in the water column (Babbin et al., 2015). This $O_2$-driven decoupling of anaerobic reactions is consistent with the observed sequential inhibition of $N_2O$ and $N_2$ production in incubation experiments (Dalsgaard et al., 2014), although we find $O_2$ inhibition thresholds that are somewhat higher than suggested by those experimental studies. Conversely, other studies have suggested much higher $O_2$ inhibition thresholds for anaerobic processes, on the order of several mmol m$^{-3}$ (Kalvelage et al., 2011; Ji et al., 2018a).

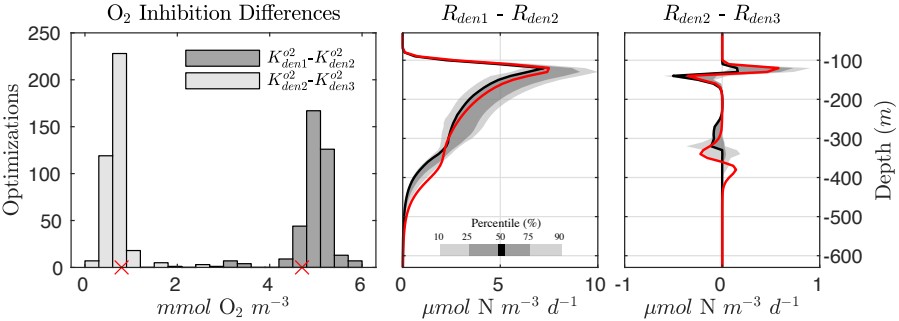

**Figure 6.** Progressive $O_2$ inhibition of denitrification steps. (left) Histogram showing the distribution for all optimized solutions of the difference in the $O_2$ inhibition constant for $NO_3^-$ and $NO_2^-$ reduction ($K_{den1}^{o2}$ and $K_{den2}^{o2}$, in dark gray), and $NO_2^-$ and $N_2O$ reduction ($K_{den2}^{o2}$ and $K_{den3}^{o2}$, in light gray). The small red markers denote the values from Opt$_{sel}$. (middle) Rate differences between $NO_3^-$ and $NO_2^-$ reduction ($R_{den1}$ and $R_{den2}$). Shading represents the 10/90 and 25/75 percentile at each vertical level from the 382 analyed parameter sets. The bold red curves denote Opt$_{sel}$ results. (right) Same as (middle), but for the difference in $NO_2^-$ and $N_2O$ reduction rates ($R_{den2}$ and $R_{den3}$).

The vertical profile of the step-wise denitrification rates ($R_{den1}$, $R_{den2}$, and $R_{den3}$) shows remarkable agreement across solutions, with only a small subset of parameter sets that behave as outliers (Fig. 5). As a consequence, the fraction of POC remineralized by each heterotrophic reaction remains consistent across optimizations (Fig. 7, top panels). Near the base of the euphotic zone, around 30 m depth, aerobic remineralization ($R_{rem}$) far exceeds denitrification, reflecting $O_2$ inhibition of the latter. However, as $O_2$ decreases to suboxic levels around 100 m depth, $NO_3^-$ reduction becomes the dominant remineralization pathway (up to 60% of total remineralization). As $O_2$ drops further within the OMZ core ($\sim$100 to 350 m depth), $NO_2^-$ and $N_2O$ reduction rapidly take up the remaining fraction ($\sim$25% and 15%, respectively), albeit with more variability than near

the euphotic zone. Below the OMZ, as the water column reverts to oxic conditions, aerobic remineralization dominates, and by 500 m depth, all solutions show essentially no denitrification.

The processes responsible for fixed N loss (anammox, $NO_2^-$ reduction, and $N_2O$ production from $NH_4^+$ oxidation) are also consistent across optimizations (Fig. 7, bottom panels). Within oxygenated waters, $N_2O$ production from $NH_4^+$ oxidation ($R_{ao}^{n2o}$) is by far the dominant fixed N loss term, as all other sources are inhibited by $O_2$. Anammox ($R_{ax}$) becomes the dominant term within the upper and lower oxycline due to increased availability of both $NO_2^-$ (from denitrification and nitrification) and $NH_4^+$ (from the decomposition of sinking POC), consistent with observations (Babbin et al., 2020). In the anoxic OMZ core,

relief from $O_2$ inhibition allows $NO_2^-$ reduction to outcompete anammox for $NO_2^-$ and contribute up to 60% of the total N loss, with anammox making up the remaining 40% (also see Fig. 5). This is somewhat higher than expected from purely stoichiometric constraints (Koeve and Kähler, 2010; Bianchi et al., 2014), likely reflecting vertical transport of $NO_2^-$ and $NH_4^+$, co-occurrence of aerobic and anaerobic processes, and the higher $O_2$ threshold for anammox inhibition in oxygenated waters. The resulting profile of total N loss thus reveals a subsurface maxima predominantly driven by anammox, with denitrification

leading total OMZ losses.

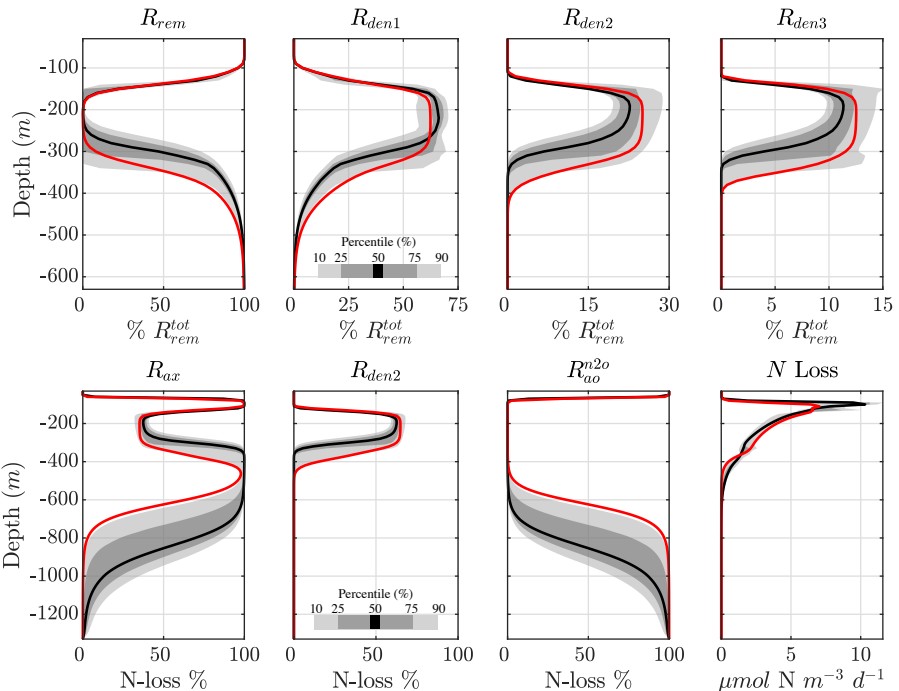

**Figure 7.** Contribution of different reactions to organic matter remineralization and fixed N loss. (top) Fraction of total POC remineralized by each heterotrophic rate ($R_{rem}$, $R_{den1}$, $R_{den2}$, and $R_{den3}$). Shading represents the 10/90 and 25/75 percentile at each vertical level from the 382 analyed parameter sets. The bold red curves denote the selected parameter set (Opt$_{sel}$) discussed in Sect. 5.2. (bottom) Same as (top), but for the fraction of total fixed N loss (via production of $N_2$ and $N_2O$) from anammox ($R_{ax}$), $NO_2^-$ reduction ($R_{den2}$), and $N_2O$ production from $NH_4^+$ oxidation ($R_{ao}^{n2o}$). The total fixed N loss is also shown. Note the different vertical axis for the bottom panels.

## 5.2 Selected solution for the Eastern Tropical South Pacific

Among tracers, $N_2O$ profiles show significant variability between optimizations. While all optimizations generate two peaks in $N_2O$ surrounding the oxygen deficient core, only a subset is able to reproduce the observed magnitude of the secondary peak at the lower oxycline (roughly 500 m depth, see Fig. 5). This subset forms a "cluster"' of optimizations that share common features that facilitate the formation of a realistic deep $N_2O$ peak, including higher $O_2$ inhibition thresholds (between 1.0 and 2.0 mmol m$^{-3}$ for $NO_2^-$ reduction, and between 0.5 and 1.0 mmol m$^{-3}$ for $N_2O$ reduction), and a wider $O_2$ window where net $N_2O$ production is favored (between 0.5 and 1.0 mmol m$^{-3}$ width). Additionally, while most optimizations are able to reproduce the OMZ peak in $NO_2^-$, significant variability in its magnitude exists. Given the central roles of $N_2O$ and $NO_2^-$ in both nitrification and denitrification pathways (Fig. 1), and the importance of oceanic $N_2O$ emissions to the atmosphere, we assign high priority to optimizations that reproduce realistic features in the distribution of these tracers, in particular a higher magnitude for the secondary $N_2O$ maximum. To this end, we select a parameter set (hereafter $Opt_{sel}$) which results in $N_2O$ and $NO_2^-$ profiles closer to observations (bold red curves in Fig. 5, with parameter values reported in Table B1). We use this $Opt_{sel}$ parameter set for further analysis of the model sensitivity.

Compared to the other parameter sets, $Opt_{sel}$ is characterized by weaker maximum $NH_4^+$ and $NO_2^-$ oxidation rates ($k_{ao}$ and $k_{no}$, respectively) and smaller half saturation constants for reductant uptake ($K_{ao}^{nh_4}$ and $K_{no}^{no_2}$, respectively) (Fig. C2). In surface oxygenated waters, this results in relatively higher $NH_4^+$ and $NO_2^-$ (Fig. 5). In contrast, maximum denitrification rates ($k_{den1}$, $k_{den2}$, and $k_{den3}$) are close to the median values from all optimizations. Rates of $NO_2^-$ and $N_2O$ reduction ($R_{den2}$ and $R_{den3}$, respectively) are generally larger than other solutions, in particular near the lower oxycline (Fig. 5). This increases POC consumption within this depth range via denitrification compared to other solutions (Fig. 7). As a consequence, the residual between the $NO_3^-$ and $NO_2^-$ reduction ($R_{den1}$ - $R_{den2}$, see Fig. 6) leads to higher $NO_2^-$ accumulation at these depths, providing the necessary $NO_2^-$ substrate to fuel either $NO_2^-$ reduction (i.e. $N_2O$ production) or anammox. Since the parameterization scheme in $Opt_{sel}$ also results in reduced $NO_2^-$ oxidation ($R_{no}$) and anammox ($R_{ax}$) rates (see Fig. 5), likely because of higher anammox half saturation constants for substrate uptake ($K_{ax}^{nh_4}$ and $K_{ax}^{no_2}$), more $NO_2^-$ is available for reduction by denitrification, leading to a surplus in production ($R_{den2}$) relative to consumption ($R_{den3}$), and high concentrations of $N_2O$ at the lower oxycline.

## 5.3 Sensitivities to model parameters

As shown in Sect. 5.1 and Fig. 4, strong correlations exist between parameter pairs in the optimization ensemble. Since $Opt_{sel}$ demonstrates good comparisons with ETSP tracer and rate observations, we perform a series of sensitivity tests around parameters ($P$) most responsible for controlling specific features ($F$) of the tracer distributions. These include concentrations of $NH_4^+$ and $NO_2^-$ at 50 m depth, the peak $NO_2^-$ concentration in the OMZ, the $N_2O$ concentrations at the primary and secondary $N_2O$ maxima, and the minimum in the OMZ $NO_3^-$ deficit (i.e., $N^*$). Additionally, we evaluate which parameters govern total N loss, including the fractional contribution of anammox, the partitioning of POC consumption via $NO_3^-$, $NO_2^-$, and $N_2O$ reduction, and total $N_2O$ production and air-sea flux (here, approximated by the vertical transport at the upper model boundary). To this

end, we calculated the sensitivity coefficient ($\phi_{ij}$) for each $P$ and $F$ pairing by evaluating the impact of varying each $\text{Opt}_{sel}$ $P$ value by $\pm 5\%$ of its range in Table B1, and recording the resulting relative change in the $F$:

$$\phi_{ij} = \frac{P_i}{F_j} \cdot \frac{\partial F_j}{\partial P_i} \tag{12}$$

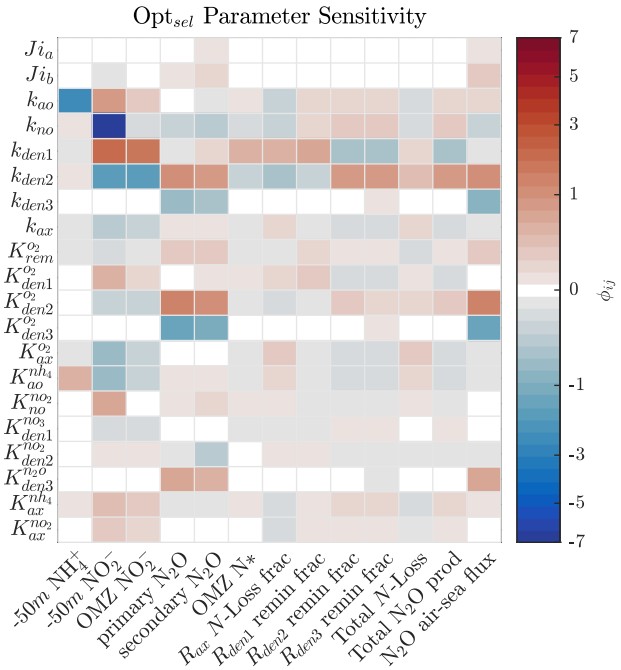

**Figure 8.** Sensitivity coefficient ($\phi_{ij} = \frac{P_i}{F_j} \cdot \frac{\partial F_j}{\partial P_i}$) for $\text{Opt}_{sel}$ parameters ($P_i$) and features ($F_j$) of the model solution. Here, each parameter is varied by $\pm 5\%$ of their respective CMA-ES allowed ranges in Table B1 to evaluate the relative impact to each feature of the model solution. Concentrations of $NH_4^+$ and $NO_2^-$ at 50 m depth (-50 m) are used as proxies of near-surface values.

The results demonstrate high sensitivity to changes in the maximum rates for all reactions (Fig. 8). Specifically, higher maximum rates correlate negatively with the concentrations of their substrates, and positively with the concentrations of their products. For example, increasing $k_{den1}$ results in an increase in OMZ $NO_2^-$, and a decrease in OMZ $N^*$. Similarly, increasing $k_{den2}$ decreases OMZ $NO_2^-$ and increases $N_2O$ concentrations in the upper and lower oxycline, and its flux to the atmosphere. These impacts are further modulated by the half saturation and $O_2$ inhibition constants.

Fig. 9 and 10 further summarize the sensitivities to the maximum denitrification rates and their inhibition by $O_2$, detailing the resulting changes to $O_2$, $N_2O$, $NO_2^-$, and $N^*$ profiles. As expected, changes in maximum rates affect reaction substrates and products in opposite ways. For example, a positive perturbation of $k_{den1}$ (top panels) stimulates $NO_3^-$ reduction, causing an increase in OMZ $NO_2^-$ and a decrease in $N^*$ as expected. Similarly, a positive perturbation of $k_{den2}$ increases $N_2O$ and decreases $NO_2^-$ nearly everywhere. However, these sensitivities have also specific depth-dependent signatures. While changes

in $NO_2^-$ are more pronounced within the OMZ core, in particular the upper section, changes in $N_2O$ are stronger at the upper and lower oxyclines, i.e., within the $N_2O$ production window defined by $K_{den2}^{o2}$ and $K_{den3}^{o2}$ (see Sect. 5.1).

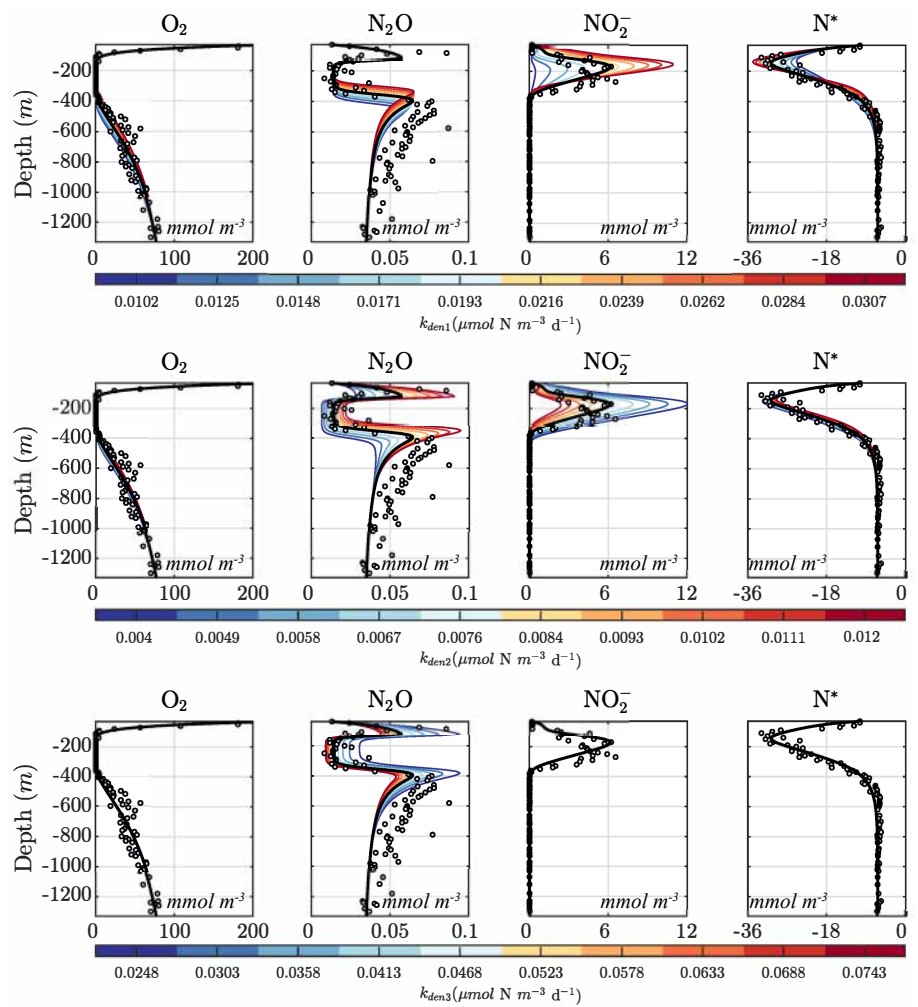

**Figure 9.** Model sensitivity to parameter values. Panels show $O_2$, $N_2O$, $NO_2^-$, and $N^*$ for the $Opt_{sel}$ parameter set after varying the maximum $NO_3^-$, $NO_2^-$, and $N_2O$ reduction rate parameters ($k_{den1}$, $k_{den2}$, and $k_{den3}$ by $\pm 5\%$ of their $Opt_{sel}$ value in Table B1.

Notably, by increasing $k_{den1}$ (top panels in Fig. 9) or $k_{den2}$ (middle panels) from $Opt_{sel}$ values, the vertical extent of oxygen
deficient waters is reduced as a result of increased POC consumption via denitrification (not shown). This enhances aerobic remineralization and nitrification below the OMZ, providing an enhanced source of $NO_3^-$ that partly offsets the OMZ losses seen via $k_{den1}$ enhancement. This may indicate a potential negative feedback: if denitrification is locally enhanced (i.e. via increased competition for POC by denitrifying heterotrophs), a resulting reduction in the vertical extent of the OMZ would inhibit further N loss.

Fig. 8 and 10 highlight significant sensitivities to the $O_2$ inhibition constants, which control $O_2$-dependent modulation of the maximum reaction rates. These effects are particularly evident at the boundaries of the OMZ. For example, an increase in $K^{o2}_{den2}$ allows for more $NO_2^-$ reduction at higher $O_2$, leading to a slight depletion in OMZ $NO_2^-$ and, as a consequence, an increase to suboxic $N_2O$ concentrations (Fig. 10, middle panels), consistent with observations of these processes in the Peruvian oxygen deficient zone (Frey et al., 2020). In a similar manner, an increase in $K^{o2}_{den3}$ leads to more $N_2O$ reduction,

reducing the magnitude of both the primary and secondary $N_2O$ peaks, while leaving other OMZ tracers ($NO_2^-$, $N^*$) relatively unaffected.

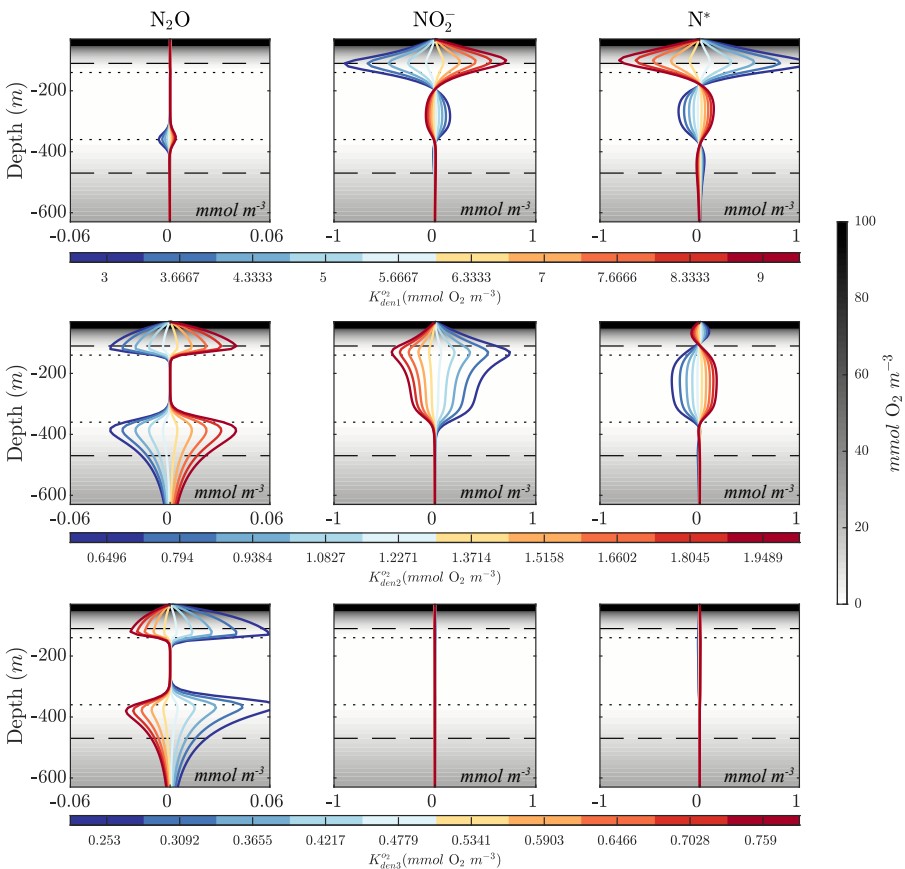

**Figure 10.** Model sensitivity to parameter values. Panels show changes to $N_2O$, $NO_2^-$, and $N^*$ for the $Opt_{sel}$ parameter set after varying the $O_2$ inhibition constants for $NO_3^-$, $NO_2^-$, and $N_2O$ reduction ($K^{o2}_{den1}$, $K^{o2}_{den2}$, and $K^{o2}_{den3}$) by $\pm 5\%$ of their $Opt_{sel}$ value in Table B1. Background gray shadings show $O_2$ concentrations, with horizontal lines higlighting $O_2$=1 mmol m$^{-3}$ (dotted lines) and $O_2$=10 mmol m$^{-3}$ (dashed lines).

## 5.4 Sensitivities to environmental variables

The main features of the OMZ simulated by the model are strongly dependent on environmental parameters such as upwelling and mixing, organic matter fluxes, and the model boundary conditions, including mixed layer depth and $O_2$ concentrations. Critically, these parameters are likely to vary over time under the effects of natural climate variability (e.g., Deutsch et al. (2011)), and anthropogenic climate change (Bopp et al., 2013). While each of these parameters control OMZ tracer profiles and N cycle reactions in complex ways, the main responses can be ascribed to changes in the position, thickness, and strength of the anoxic OMZ layer. Perturbations that replenish $O_2$ above the thresholds for anoxic processes — such as those predicted under climate warming scenarios (Busecke et al., 2022) — have thus cascading impacts on anaerobic N cycle intermediates such as $NO_2^-$ and $N_2O$, and on the fixed N removal and $NO_3^-$ deficit of the oxygen deficient zone.

Fig. 11 shows the sensitivity of the optimal solution $Opt_{sel}$ to the magnitudes of vertical upwelling ($w_{up}$) and turbulent diffusion ($K_v$). Increasing $w_{up}$ results in higher $O_2$ supply from below the OMZ, leading to increasing $O_2$ concentrations, and an upward shift and thinning of the anoxic layers. At high upwelling, the anoxic layer is effectively wiped out and is replaced by a suboxic layer. Similar results are obtained with higher $K_v$ values, with an increase of diffusive $O_2$ supply from both above and below the OMZ, resulting in a progressive shrinking of the anoxic layer. As this layer vanishes, anaerobic processes cease, drastically reducing the concentration of $NO_2^-$ and the N deficit in the OMZ core. Notably, as the OMZ reaches the brink of anoxia, i.e., as the minimum $O_2$ concentration falls within the $N_2O$ production window, the upper and lower $N_2O$ maxima merge into a single $N_2O$ spike with particularly high $N_2O$ concentrations, reflecting the largest imbalance between production and consumption.

Opposite changes are observed for a reduction in both $w_{up}$ and $K_v$, which result in an expansion of the OMZ layer, increased $NO_3^-$, $NO_2^-$, and $N_2O$ reduction, a larger OMZ $NO_2^-$ peak, and a broader separation of the upper and lower $N_2O$ maxima. The interplay between the position of the oxygen deficient layer, sinking particle fluxes, and transport processes further modulates the response of tracer profiles. For example, as anoxic waters expand upwards following a reduction in $K_v$, they intercept a higher concentration of sinking organic matter, which in turn fuels higher remineralization rates. Together with reduction in diffusive fluxes, this likely favors the strengthening of the upper $N_2O$ maximum at low $K_v$ observed in Fig. 11.

Because the supply of POC to the OMZ controls the overall magnitude of remineralization reactions, including $O_2$ consumption and denitrification, the model is particularly sensitive to the sinking POC flux at the upper model boundary ($\Phi_{poc}^{top}$, Table B3; Fig. 12, top panel). Increasing $\Phi_{poc}^{top}$ causes a greater remineralization rate, which reduces available $O_2$, and drives a progressive thickening of the OMZ, with a series of cascading impacts on tracers similar to the ones discussed above. In contrast, decreasing $\Phi_{poc}^{top}$ reduces the $R_{rem}$ rates and increases $O_2$ to the point that anoxic conditions and their signature disappear.

Similar changes can also be driven by variations in the bottom boundary $O_2$ concentration, which directly controls upward $O_2$ supply by upwelling (Fig. 12, bottom panel). Increasing bottom $O_2$ progressively decreases the thickness of the OMZ, shifting it upwards, and eventually eroding the anoxic layer. Conversely, decreasing bottom $O_2$ leads to a downward expansion of the OMZ and an intensification of anoxic conditions and the resulting anaerobic reactions.

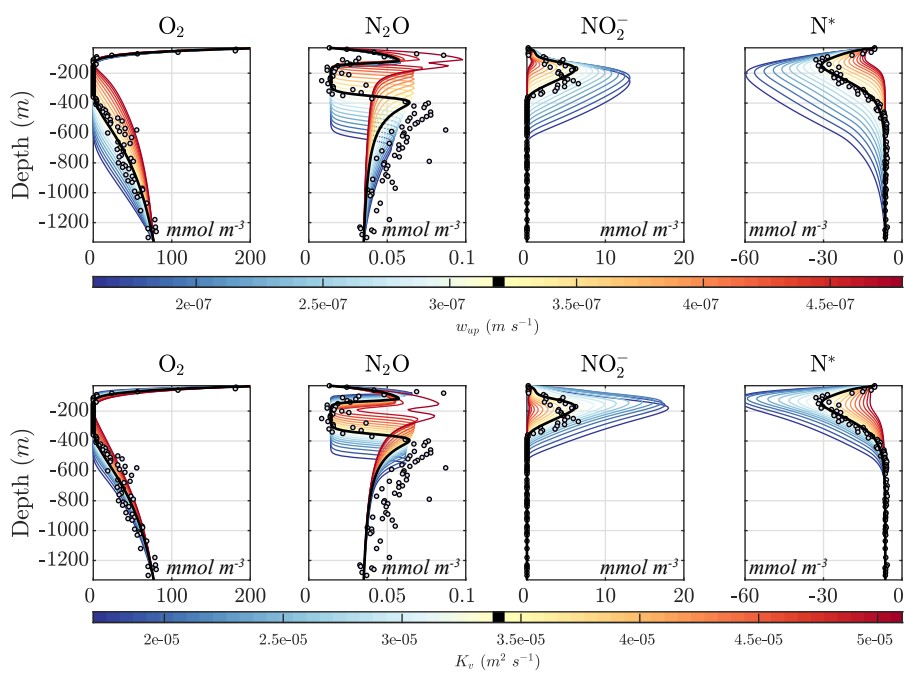

**Figure 11.** Model sensitivity to physical drivers. (top) Sensitivity of the Opt$_{sel}$ optimized solution to the constant vertical upwelling velocity ($w_{up}$). (bottom) Sensitivity to the vertical turbulent diffusion coefficient ($K_v$). The bold black curves indicate original Opt$_{sel}$ values, which are also indicated on their respective colorbars.

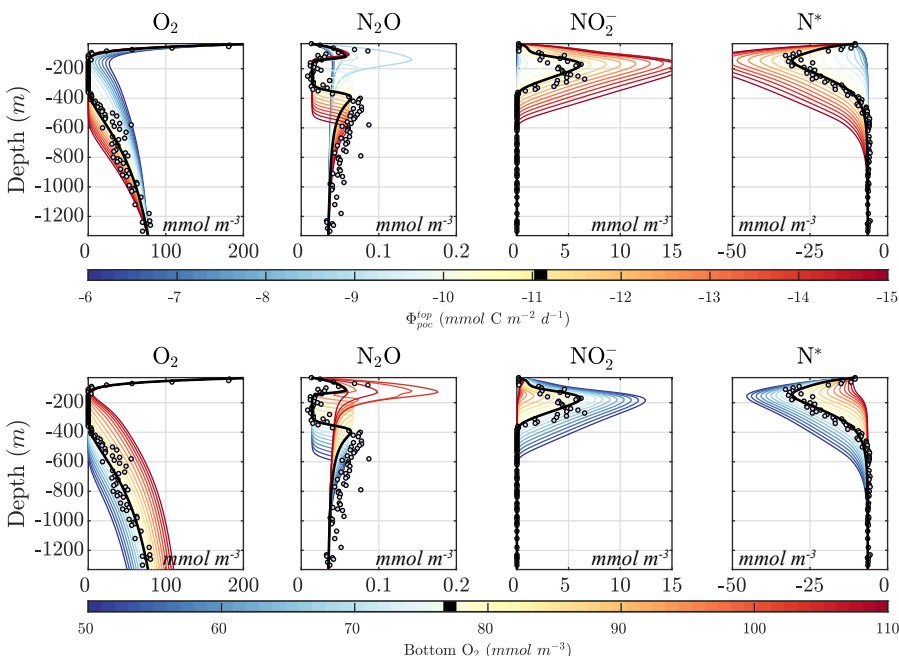

**Figure 12.** Model sensitivity to biogeochemical drivers. Same as in Fig. 11, but for surface POC flux ($\Phi_{poc}^{top}$) and $O_2$ concentration at the lower model boundary. In the top panels, more negative values of $\Phi_{poc}^{top}$ correspond to an increasing sinking POC flux.

## 6 Discussion and conclusions

We developed a model of the N cycle in low $O_2$ waters, and optimized it to reproduce observations from the ETSP OMZ. The model is able to simulate the distribution of multiple N cycle tracers, including $NO_2^-$ and $N_2O$, and their transformation rates, capturing the underlying dynamics and environmental sensitivity of the underlying reactions (Fig. 5). In general, the model reproduces observed tracer concentration profiles more accuratetely than transformation rates. Mismatches with transformation rates may point to processes that need improvement in the model, but also underscore limitations in rate measurements, which rely on shipboard incubation experiments that are usually more uncertain and limited than tracer measurements, and may not perfectly reflect in situ conditions. However, by matching observed reaction rates to a reasonable degree, the model approximates the complex dynamics of the system in a way that allows it to reproduce tracer distributions. Co-located tracer and rate measurements for multiple processes are thus an effective way to constrain the model representation of the N cycle in and around $O_2$-deficient environments.

The optimization indicates that multiple parameter sets can produce equally good fits to tracer and rate profiles (Fig. 5). This is expected given the non-linear nature of the model, and limitations in the observations. Even when rate measurements are used to constrain the model, as done here, an ensemble of equally good solutions is thus possible. This optimized ensemble shows that significant variability and trade-offs can exist between specific parameters (Fig. 4), suggesting that compensation between different procesess can lead to similar profiles of tracers and transformation rates. Refinements to the criteria used to optimize the model, i.e., additional constraints in the definition of the cost function, could allow to further narrow down plausible sets of parameters. For example, to evaluate the model sensitivity (Fig. 8-10), we select a parameter set from our optimization ensemble that better captures the magnitude of the secondary $N_2O$ maximum, while reproducing other observed features equally well. While we adopt a relatively simple cost function definition, additional constraints such as this one could be explicitly built into its formulation and weighted more heavily to revise model parameters.

A better characterization of environmental sensitivities to substrate concentrations (e.g., half saturation constant for substrate uptake) and $O_2$ sensitivities would also help parameter selection, for example by narrowing down the prior and posterior range of values for these and other variables (e.g., maximum reaction rates). To this end, rate measurements under a range of $O_2$ and substrate concentrations are especially helpful. Similarly, simultaneous optimization of the model to reproduce observations across multiple regions of an OMZ characterized by different conditions, e.g., the core and the boundaries, or across different OMZ and oceanographic regimes, would likely result in more robust optimizations.

Despite the variability in parameter values, analysis of the optimal ensemble reveals emerging features that appear robust across multiple optimizations and that compare well with observations. For example, the sensitivity of denitrification processes to $O_2$ shows systematic variations, with weaker $O_2$ inhibition for $NO_3^-$ reduction, and stronger for $N_2O$ reduction (Fig. 6). Accordingly, $NO_3^-$ reduction to $NO_2^-$ tends to occur at higher $O_2$ concentrations than $NO_2^-$ reduction to $N_2O$, which in turns occurs at higher $O_2$ concentrations than $N_2O$ reduction to $N_2$. This result is consistent with tracer incubation experiments (Dalsgaard et al., 2014). However, we note that the specific value of these $O_2$ sensitivities is far from well-established, with some experiments showing smaller thresholds than found in our optimization (Dalsgaard et al., 2014), and others finding

similar or larger thresholds (Ji et al., 2018a). In the model, the sequential sensitivity of denitrification steps to $O_2$ supports an $O_2$-dependent window for $N_2O$ production, which allows accumulation of $N_2O$ at the margins of the OMZ core. This, and other systematic relationships between parameters and features of the solutions, as revealed by a sensitivity analysis (Fig. 8-10), shed light on specific balances in the N cycle, and can be exploited as a powerful tool to fine-tune the model, both in the 1D setup used here, and in more complex and resource-intensive 3D implementations where a formal optimization would be

unfeasible (McCoy et al., 2022).

Because the model is based on a mechanistic representation of N transformations, it is suitable to investigate the response of the N cycle to environmental variability and other perturbations (Fig. 11-12). For example, the model could be used to investigate the effects of eddy variability near the boundaries of OMZs, or the effects of OMZ expansion and change under global warming. With these goals in mind, the model is designed to be coupled to the biogeochemical component of the current

generation of Earth System Models, enabling accurate simulation of $NO_2^-$ and $N_2O$ dynamics, with minimal interference with the representation of the cycles of oxygen, nutrients, carbon, and organic matter.

Because the model reflects an evolving understanding of the N cycle, its assumptions should be re-evaluated as new N transformation processes and aspects of microbial dynamics are uncovered. The model is built around two major simplifications: the "modularity'" of the N cycle, and the representation of microbial metabolisms as "bulk'" chemical reactions that avoid

explicitly tracking diverse microbial populations. Both are approximate views of the N cycle. For example, recent evidence suggests that microorganisms with the ability to carry out intracellular reduction of $NO_3^-$ to $NO_2^-$ and $NO_2^-$ to $N_2O$ may dominate production of $N_2O$ in oxygen deficient waters (Ji et al., 2018a; Frey et al., 2020), although the sensitivity of this process to environmental factors is still being uncovered.

Our "bulk'" approach assumes that metabolic reaction rates are proportional to substrates following a Michaelis-Menten

dependency. However, in reality, reaction rates also depend on the abundance of microorganisms present in the water column. If microorganism biomass is assumed to be proportional to substrates, then a higher-order dependency of reaction rates may be more appropriate, as adopted by some biogeochemical models (e.g., Paulot et al. (2020)). A different dependence on substrates, in turn, may affect the variability of reaction rates with depth, and the model sensitivity to parameters such as maximum reaction rates.

Indeed, previous modeling studies have pointed out the value of explicitly resolving the biomass of microbial populations (Penn et al., 2016; Zakem et al., 2020). This, in turn, enables a more direct comparison of model results with molecular observations (Louca et al., 2016), and would favor the emergence of complex feedbacks between microbes and their substrates driven by resource competition and oceanic circulation (Penn et al., 2019). However, explicitly simulating microbial biomass requires a number of additional parameters that remain poorly constrained and adds computational burden that may not always

improve the realism of biogeochemical simulations (Galbraith et al., 2015). Our model provides a valuable framework to continue exploring these ideas in both idealized and realistic settings (McCoy et al., 2022).

Based on its modular design, the model can be naturally expanded to represent new processes that, while thought to be relevant in OMZ, are still uncertain. These include: (1) Additional known N cycle pathways, and their sensitivity to environmental variability, such as DNRA (Lam et al., 2009), hybrid $N_2O$ production from AOA (Stieglmeier et al., 2014), and direct $NO_3^-$

reduction to $N_2O$ (Ji et al., 2018a; Frey et al., 2020); (2) Alternative oxidation pathways, for example $NO_2^-$ oxidation with iodate, or $NO_2^-$ disproportionation reactions (Babbin et al., 2020; Buchwald et al., 2015a; Sun et al., 2021a); (3) Coupling of N tracers with the cycles of other elements, e.g. carbon, sulfur and iron, such as chemolithotrophic denitrification coupled to hydrogen sulfide ($H_2S$) oxidation, or anaerobic $NO_2^-$-based methane ($CH_4$) oxidation (Azhar et al., 2014; Scholz et al., 2016; Thamdrup et al., 2019; Callbeck et al., 2021); (4) Explict representation of chemolithrotrophy and its effects on organic matter

fixation (Swan et al., 2011); (5) Explicit coupling to the inorganic carbon cycle, by inclusion of $CO_2$ and alkalinity changes associated with N cycle reactions (Cinay et al., 2022); (6) The cycling of nitric oxide (NO) (Ward and Zafiriou, 1988; Lutterbeck et al., 2018); (7) A more detailed representation of the microbial ecology underlying the N cycle (Louca et al., 2016; Zakem et al., 2018; Penn et al., 2019).

*Code and data availability.* The current version of NitrOMZv1.0 is available from the project website: https://doi.org/10.5281/zenodo.7106213.

The exact version of the model used to produce the results used in this paper is archived on Zenodo (Daniele Bianchi, Simon Yang, and Daniel McCoy. (2022). NitrOMZv1.0 Model Code (1.0). Zenodo. https://doi.org/10.5281/zenodo.7106213), as are input data and scripts to run the model and produce the plots for all the simulations presented in this paper (Bianchi et al., 2022).

*Author contributions.* DB conceptualized the formulation of the model; DB and SY developed the model code, including optimization procedures; DB and SY organized the validation data; DB and DM designed the analyses; DM prepared the tables and visualization of data;

DB and DM prepared the manuscript with contributions from SY.

*Competing interests.* The authors declare that they have no conflict of interest.

*Acknowledgements.* This material is based upon work supported by the U.S. National Science Foundation under grant OCE-1847687. D.B. acknowledges support from the Alfred P. Sloan Foundation, and computational support by the Extreme Science and Engineering Discovery Environment (XSEDE) through allocation TG-OCE17001. The Authors wish to thank A. Babbin, A. Santoro, C. Kelly for helpful discussion.

## Appendix A: NitrOMZ Equations

### A1 Heterotrophic rate equations

$$R_{rem} = k_{rem} \cdot \frac{O_2}{K_{rem}^{o2} + O_2} \cdot POC \tag{A1}$$

$$R_{den1} = k_{den1} \cdot \frac{NO_3^-}{K_{den1}^{no3} + NO_3^-} \cdot e^{\frac{-O_2}{K_{den1}^{o2}}} \cdot POC \tag{A2}$$

$$R_{den2} = k_{den2} \cdot \frac{NO_2^-}{K_{den2}^{no2} + NO_2^-} \cdot e^{\frac{-O_2}{K_{den2}^{o2}}} \cdot POC \tag{A3}$$

$$R_{den3} = k_{den3} \cdot \frac{N_2O}{K_{den3}^{n2o} + N_2O} \cdot e^{\frac{-O_2}{K_{den3}^{o2}}} \cdot POC \tag{A4}$$

### A2 Chemolithotrophic rate equations

$$R_{ao} = k_{ao} \cdot \frac{O_2}{K_{ao}^{o2} + O_2} \cdot \frac{NH_4^+}{K_{ao}^{nh4} + NH_4^+} \tag{A5}$$

$$R_{no} = k_{no} \cdot \frac{O_2}{K_{no}^{o2} + O_2} \cdot \frac{NO_2^-}{K_{no}^{no2} + NO_2^-} \tag{A6}$$

$$R_{ax} = k_{ax} \cdot \frac{NH_4^+}{K_{ax}^{nh4} + NH_4^+} \cdot \frac{NO_2^-}{K_{ax}^{no2} + NO_2^-} \cdot e^{\frac{-O_2}{K_{ax}^{o2}}} \tag{A7}$$

### A3 Aerobic N$_2$O production

Production of N$_2$O via the nitrification pathway in NitrOMZ (pathway 2b in Fig. 1) is modelled as a by-product of $R_{ao}$ with enhanced yields at lower O$_2$ concentrations. The partitioning between N$_2$O and NO$_2^-$ production from $R_{ao}$ is calculated using the function proposed by Nevison et al. (2003), which was derived by fitting measured N$_2$O and NO$_2^-$ yields ($Y_{ao}^{n2o}$ and $Y_{ao}^{no2}$, respectively) to oxygen concentrations (Goreau et al., 1980), and re-fit by multiple observations in the Eastern Tropical North and South Pacific OMZ (Ji et al., 2015a, 2018a; Santoro et al., 2021):

$$\frac{Y_{ao}^{n2o}}{Y_{ao}^{no2}} = \left( \frac{Ji_a}{[O_2]} + Ji_b \right) \cdot 0.01 \tag{A8}$$

Nitrification-derived NO$_2^-$ and N$_2$O production rates ($R_{ao}^{no2}$ and $R_{ao}^{n2o}$ respectively, pathways 2a and 2b in Fig. 1) are therefore represented as:

$$R_{ao}^{n2o} = R_{ao} \cdot Y_{ao}^{n2o}, \tag{A9}$$

$$R_{ao}^{no2} = R_{ao} \cdot Y_{ao}^{no2}. \tag{A10}$$

### A4 Stoichiometry

The stoichiometry of heterotrophic redox reactions is based on an electron balance and follows the procedure outlined in Paulmier et al. (2009), under the assumption that the composition of organic matter (POC) follows the average oceanic ratios

from Anderson and Sarmiento (1994): $C_{106}H_{175}O_{42}N_{16}P$. This chemical composition can be arbitrarily adjusted in NitrOMZ. For example, studies in the Eastern Tropical South Pacific suggest a C:N ratio closer to 83:1 (Teng et al., 2014). Furthermore, organic matter degradation reactions may also differentially remineralize C, N and P. For instance, denitrification may preferentially involve degradation of amino-acids, and thus impact the N:P ratio of remineralization differently from aerobic respiration (Van Mooy et al., 2002).

Based on the stoichiometry of Anderson and Sarmiento (1994), the $NH_4^+$ to C and $PO_4^{3-}$ to C ratios during aerobic respiration are:

$$Q_{rem}^{N:C} = \frac{16}{106}, \tag{A11}$$

$$Q_{rem}^{P:C} = \frac{1}{106}. \tag{A12}$$

As a result of the POC composition, a total of 472 electrons are required to oxidize POC to $CO_2$. With 4 electrons required to reduce $O_2$ to $H_2O$, the oxygen to carbon remineralization ratio for aerobic remineralization to $NH_4^+$ is represented as:

$$Q_{rem}^{O:C} = \frac{472}{106 \cdot 4}, \tag{A13}$$

This yields a respiration quotient of $r_{-O_2:C}$ of 1.11, which is within range of direct chemical measurements of $r_{-O_2:C}$ from Moreno et al. (2020, 2022). For nitrification, the oxygen to nitrogen ratios for $NH_4^+$ and $NO_2^-$ oxidation ($R_{ao}$ and $R_{no}$, respectively) are based on the stoichiometry of the relevant redox reactions:

$$Q_{ao}^{O:N} = \frac{3}{2}, \tag{A14}$$

$$Q_{no}^{O:N} = \frac{1}{2}. \tag{A15}$$

For denitrification, 2 electrons are required for each respective reduction step ($NO_3^-$ to $NO_2^-$, $NO_2^-$ to $\frac{1}{2}N_2O$, and $N_2O$ to $N_2$), thus the corresponding ratios are:

$$Q_{den}^{N:C} = \frac{472}{106 \cdot 2} \tag{A16}$$

Finally, for anammox, $NH_4^+$ and $NO_2^-$ are combined in 1:1 ratios to produce $N_2$. The above ratios are then applied to the tracer equations in Appendix A5.

## A5 Tracer sources-minus-sinks equations

$$\frac{d[\text{POC}]}{dt} = -(R_{rem} + R_{den1} + R_{den2} + R_{den3}) \tag{A17}$$

$$\frac{d[\text{O}_2]}{dt} = (Q_{rem}^{O:C} \cdot R_{rem}) - (Q_{ao}^{O:N} \cdot R_{ao}) - (Q_{no}^{O:N} \cdot R_{no}) \tag{A18}$$

$$\frac{d[\text{NO}_3^-]}{dt} = R_{no} - (Q_{den}^{N:C} \cdot R_{den1}) \tag{A19}$$

$$\frac{d[\text{PO}_4^{3-}]}{dt} = Q_{rem}^{P:C} \cdot (R_{rem} + R_{den1} + R_{den2} + R_{den3}) \tag{A20}$$

$$\frac{d[\text{NH}_4^+]}{dt} = Q_{rem}^{N:C} \cdot (R_{rem} + R_{den1} + R_{den2} + R_{den3}) - (R_{ao} + R_{ax}) \tag{A21}$$

$$\frac{d[\text{NO}_2^-]}{dt} = R_{ao}^{no_2} + Q_{den}^{N:C} \cdot (R_{den1} - R_{den2}) - (R_{no} + R_{ax}) \tag{A22}$$

$$\frac{d[\text{N}_2\text{O}]}{dt} = 0.5 \cdot (R_{ao}^{n_2o} + Q_{den}^{N:C} \cdot R_{den2}) - (Q_{N:C}^{den} \cdot R_{den3}) \tag{A23}$$

$$\frac{d[\text{N}_2]}{dt} = (Q_{den}^{N:C} \cdot R_{den3}) + R_{ax} \tag{A24}$$

$$\tag{A25}$$

## Appendix B: NitrOMZ Parameters and Configurations

**Table B1.** NitrOMZ nitrogen cycle parameters and CMA-ES optimization ranges.

| Parameter | Description | Units | CMA-ES Min Value | CMA-ES Max Value | $\text{Opt}_{sel}$ ETSP |
|---|---|---|---|---|---|
| $Ji_a$ | Nevison et al. (2003) 'a' parameter | N/A | 0.0500 | 0.4000 | 0.4000 |
| $Ji_b$ | Nevison et al. (2003) 'b' parameter | N/A | 0.0500 | 0.2000 | 0.2000 |
| $k_{rem}$ | Maximum aerobic respiration rate | $\text{d}^{-1}$ | N/A | N/A | 0.0800 |
| $k_{den1}$ | Maximum $NO_3^-$ reduction rate | $\text{d}^{-1}$ | 0.0080 | 0.0800 | 0.0205 |
| $k_{den2}$ | Maximum $NO_2^-$ reduction rate | $\text{d}^{-1}$ | 0.0080 | 0.0800 | 0.0080 |
| $k_{den3}$ | Maximum $N_2O$ reduction rate | $\text{d}^{-1}$ | 0.0080 | 0.0800 | 0.0496 |
| $k_{ao}$ | Maximum $NH_4^+$ oxidation rate | $\mu\text{mol N m}^{-3}\,\text{d}^{-1}$ | 0.0100 | 0.5000 | 0.0167 |
| $k_{no}$ | Maximum $NO_2^-$ oxidation rate | $\mu\text{mol N m}^{-3}\,\text{d}^{-1}$ | 0.0100 | 0.5000 | 0.0118 |
| $k_{ax}$ | Maximum anammox rate | $\mu\text{mol N m}^{-3}\,\text{d}^{-1}$ | 0.0100 | 0.5000 | 0.4411 |
| $K_{rem}^{o2}$ | $O_2$ half saturation constant for aerobic respiration | $\text{mmol O}_2\,\text{m}^{-3}$ | 0.0100 | 1.0000 | 1.0000 |
| $K_{ao}^{nh4}$ | $NH_4^+$ half saturation constant for $NH_4^+$ oxidation | $\text{mmol N m}^{-3}$ | 0.0100 | 1.0000 | 0.5091 |
| $K_{ao}^{o2}$ | $O_2$ half saturation constant for $NH_4^+$ oxidation | $\text{mmol O}_2\,\text{m}^{-3}$ | N/A | N/A | 0.3300 |
| $K_{no}^{no2}$ | $NO_2^-$ half saturation constant for $NO_2^-$ oxidation | $\text{mmol N m}^{-3}$ | 0.0100 | 1.0000 | 0.3053 |
| $K_{no}^{o2}$ | $O_2$ half saturation constant for $NO_2^-$ oxidation | $\text{mmol O}_2\,\text{m}^{-3}$ | N/A | N/A | 0.7780 |
| $K_{den1}^{no3}$ | $NO_3^-$ half saturation constant for $NO_3^-$ reduction | $\text{mmol N m}^{-3}$ | 0.0100 | 1.0000 | 1.0000 |
| $K_{den2}^{no2}$ | $NO_2^-$ half saturation constant for $NO_2^-$ reduction | $\text{mmol N m}^{-3}$ | 0.0100 | 1.0000 | 0.0100 |
| $K_{den3}^{n2o}$ | $N_2O$ half saturation constant for $N_2O$ reduction | $\text{mmol N m}^{-3}$ | 0.0100 | 0.2000 | 0.1587 |
| $K_{ax}^{nh4}$ | $NH_4^+$ half saturation constant for anammox | $\text{mmol N m}^{-3}$ | 0.1000 | 1.0000 | 1.0000 |
| $K_{ax}^{no2}$ | $NO_2^-$ half saturation constant for anammox | $\text{mmol N m}^{-3}$ | 0.1000 | 1.0000 | 1.0000 |
| $K_{den1}^{o2}$ | $O_2$ exponential inhibition for $NO_3^-$ reduction | $\text{mmol O}_2\,\text{m}^{-3}$ | 0.0100 | 6.0000 | 6.0000 |
| $K_{den2}^{o2}$ | $O_2$ exponential inhibition for $NO_2^-$ reduction | $\text{mmol O}_2\,\text{m}^{-3}$ | 0.0100 | 3.0000 | 1.2993 |
| $K_{den3}^{o2}$ | $O_2$ exponential inhibition for $N_2O$ reduction | $\text{mmol O}_2\,\text{m}^{-3}$ | 0.0100 | 3.0000 | 0.5060 |
| $K_{ax}^{o2}$ | $O_2$ exponential inhibition for anammox | $\text{mmol O}_2\,\text{m}^{-3}$ | 0.5000 | 6.0000 | 6.0000 |

**Table B2.** ETSP configuration for optimization routines.

| Name | Description | Unit | Value |
|---|---|---|---|
| $z^{top}$ | Minumum model depth | m | -30 |
| $z^{bot}$ | Maximum model depth | m | -1330 |
| $w_{up}$ | Upwelling velocity (constant) | $m\,y^{-1}$ | 10.0562 |
| $K_v^{top}$ | Upper boundary vertical diffusion coefficient | $m^2\,y^{-1}$ | 750.9983 |
| $K_v^{bot}$ | Lower boundary vertical diffusion coefficient | $m^2\,y^{-1}$ | 1072.8547 |
| $K_v^{flex}$ | Variable (sigmoidal) vertical diffusion parameter | m | -250 |
| $K_v^{width}$ | Variable (sigmoidal) vertical diffusion parameter | m | 300 |
| $\Phi^{top}$ | Initial POC flux @ -30m | $mmol\,C\,m^{-2}\,d^{-1}$ | -11.1 |
| $k_{rem}$ | Maximum aerobic remineralization rate | $d^{-1}$ | 0.0800 |
| $b$ | Martin coefficient | N/A | -0.7049 |

**Table B3.** ETSP boundary conditions.

| Tracer | Units | Top | Bottom |
|---|---|---|---|
| $O_2$ | $mmol\,m^{-3}$ | 225.00 | 77.00 |
| $NO_3^-$ | $mmol\,m^{-3}$ | 2.81 | 42.50 |
| $PO_4^{3-}$ | $mmol\,m^{-3}$ | 0.82 | 3.06 |
| $N_2O$ | $\mu mol\,m^{-3}$ | 13.00 | 35.00 |
| $NO_2^-$ | $mmol\,m^{-3}$ | 0.15 | 0.00 |
| $NH_4^+$ | $mmol\,m^{-3}$ | 0.40 | 0.00 |
| $N_2$ | $mmol\,m^{-3}$ | 2.00 | 6.00 |

**Table B4.** Optimized ETSP parameter sets.

| Name | Parameter Sets | # of Iterations | Perturbations | Tracer Weights ($O_2$, $NO_3^-$, $PO_4^{3-}$, $N_2O$, $NH_4^+$, $NO_2^-$, $N^*$) | Rate Weights ($R_{ao}^{n2o}$, $R_{den1}$, $R_{den2}$, $R_{ax}$) |
|---|---|---|---|---|---|
| optVK$_v$-v6 | 45 | 40k | 20% | 2, 1, 1, 6, 2, 4, 4 | 1, 1, 1, 1 |
| optVK$_v$-v7 | 69 | 40k | 20% | 2, 1, 1, 8, 0, 4, 4 | 1, 1, 1, 1 |
| optVK$_v$-v8 | 110 | 40k | 20% | 2, 1, 1, 12, 2, 4, 4 | 1, 1, 1, 1 |
| optVK$_v$-v9 | 158 | 40k | 20% | 2, 1, 1, 12, 2, 8, 4 | 1, 1, 1, 1 |

## Appendix C: Supplemental Figures

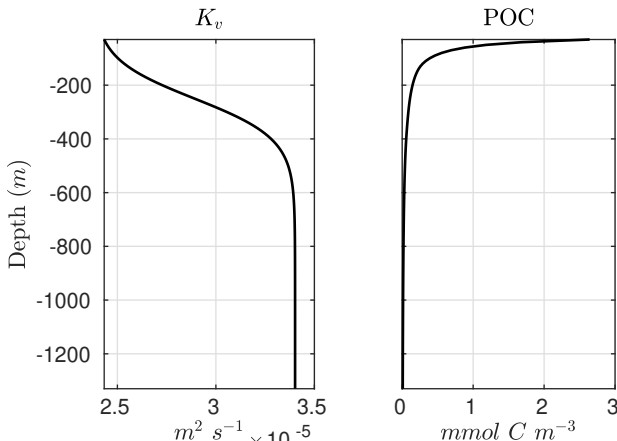

**Figure C1.** (left) Opt$_{sel}$ vertical diffusion ($K_v$) and (right) POC profiles.

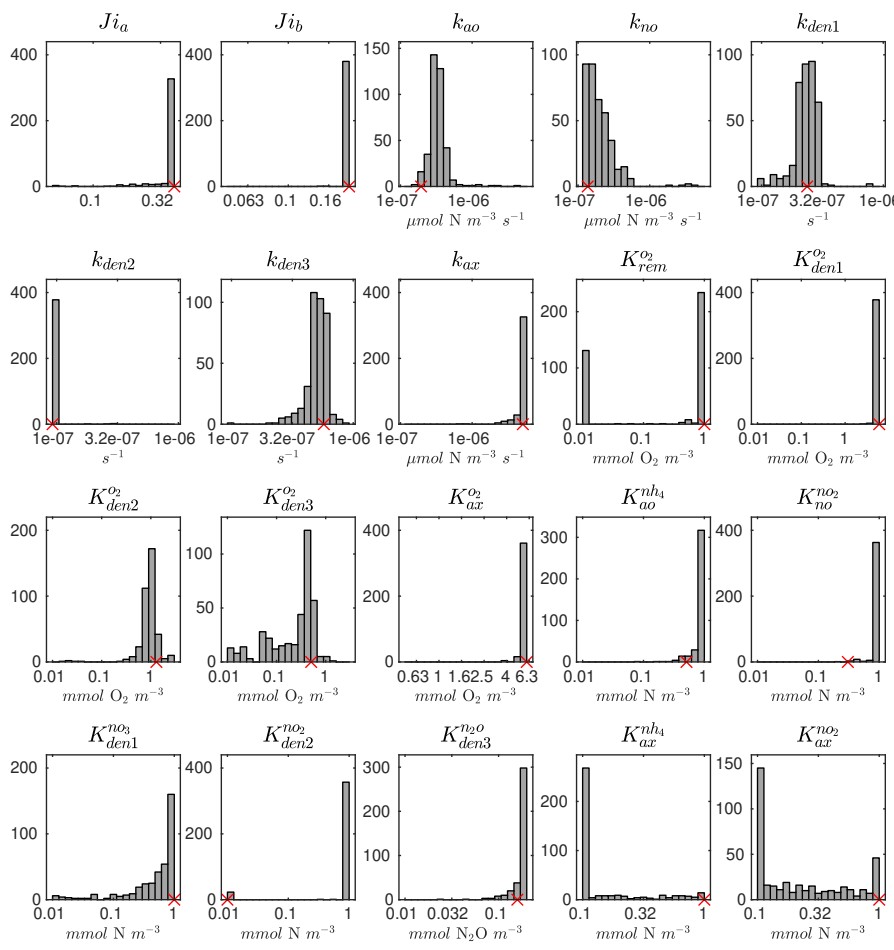

**Figure C2.** Parameter distributions from the 382 CMA-ES optimized ETSP solutions. Red markers denote $Opt_{sel}$ values.

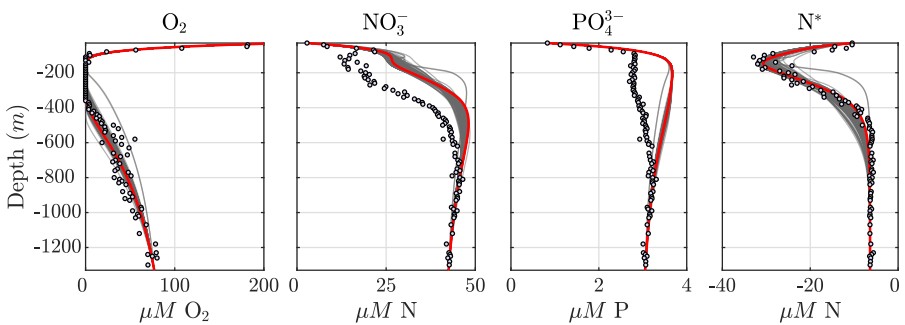

**Figure C3.** Same as in Figure 5, but for $O_2$, $NO_3^-$, $PO_4^{3-}$, and N∗.

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
