# Peer review of "Formulation, optimization and sensitivity of NitrOMZv1.0, a biogeochemical model of the nitrogen cycle in oceanic oxygen minimum zones"

_Geoscientific Model Development, 2022_

## Author Comment (AC1)

**Reviewer #1**

**Summary and general appraisal**

The manuscript titled "Formulation, optimization and sensitivity of NitrOMZv1.0, a biogeochemical model of the nitrogen cycle in oceanic oxygen minimum zones" by Bianchi et al. presented a new model of the marine nitrogen (N) cycle in and around oxygen minimum zones, which can be incorporated into current ocean biogeochemical models. This model includes the major forms of nitrogen as prognostic tracers and parameterizes their transformations as a function of organic matter availability, substrate concentrations, and the concentration of dissolved  $O_2$ . The model is formulated in an idealized 1D representation that allows for formal optimization against in situ observations and sensitivity tests to model parameters and environmental conditions.

The authors found that multiple optimal parameter sets can reproduce observed tracer and rate profiles, given the nonlinearity of the optimization problem, and trade-offs between different model parameters. Despite this variability, the authors found some consistent conclusions:

 $\cdot$  Systematic relationships exist between parameters and features of the solutions, with cascading impacts of model parameters and environmental conditions on tracer concentrations in and around the oxygen deficient region;

 $\cdot$  Different steps of the denitrification pathway had different O2 sensitivities, with increasing inhibitions for the different steps, creating an O2-dependent production window for N2

The model presented in this manuscript represents a significant improvement over other dynamical N cycle models in terms of the complexity and accuracy of its N cycle representation. This is especially important in oxygen-deficient regions such as the Eastern Tropical South Pacific, where multiple aerobic and anaerobic N cycle metabolisms may overlap. In this regard, the manuscript represents an important and valuable contribution to N cycle modeling.

We thank the reviewer for the thorough and thoughtful comments, and for the positive assessment of the manuscript.

The strengths of this paper lie in the development of the model itself, the thorough (and very useful) explanation of model equations and parameters, and the explanation of the numerical trade-offs between processes that lead to multiple optimal solutions. In addition, the clever use of an evolutionary optimization algorithm allows for the optimization of the complex and nonlinear cost function. Where the paper falls short is in its discussion of model results. There are interesting features of the solutions and sensitivity tests that bear more discussion than what the authors have provided. For example, the O2 range defined by  $K^{O2}_{den2}$  and  $K^{O2}_{den3}$  allows for N2O production at higher levels of O2 than previously thought (Dalsgaard et al. 2014). This is an interesting result and should be further discussed. Likewise, increasing the rate of nitrite reduction to N2O leads to an increase in

 $N^*$  — it is not immediately obvious why this would be the case. These and other results should be discussed in more detail, both with regards to model mechanics and in terms of their biogeochemical relevance.

These are excellent suggestions. While the manuscript focuses on the model development aspect of the study, we now strive to expand the discussion section with a broader range of findings that are relevant to ocean biogeochemistry.

My other major concern is that the authors do not discuss processes potentially missing from the model until the very last paragraph of the manuscript. In my opinion, this discussion should be moved up to Section 2, "Nitrogen cycle model formulation." To this list the authors should add dissimilatory nitrate reduction to ammonium (DNRA) (Lam and Kuypers 2011; Kraft et al. 2011), hybrid N2O production by ammonia-oxidizing archaea (Stieglmeier et al. 2014; Kozlowski et al. 2016; Trimmer et al. 2016), and N2O production directly from nitrate vie denitrification, i.e. nitrate reduction to nitrite and N2O entirely within the cell (Ji et al. 2015, 2018). The latter has been shown to occur at rates orders of magnitude higher than nitrite reduction to N2O in the eastern tropical South Pacific and eastern tropical North Pacific oxygen minimum zones (Ji et al. 2018; Frey et al. 2020) and is probably the most important omission from the model's N cycle formulation.

Thank you for the suggestion and the opportunity to clarify the philosophy of our modeling exercise. Indeed, the marine N cycle is more complex than can easily be encapsulated in an "intermediate complexity" biogeochemical model that can be embedded into current ocean circulation models. We now clarified aspects of our approach in Section 2. First, when possible, we explicitly model well-established N cycle processes under the assumption of a "modular" N cycle, whereby microbial specialization translates into separate steps for N cycle transformations. This is the rationale for modeling denitrification as three separate steps, and thus separate NO2- from N2O reduction. Second, we focus on metabolic pathways (i.e., in the "modular" N cycle case, specific reactions) that have been shown to be quantitatively important in the marine environment and for which more extensive measurements exist. This is the rationale for not including water-column DNRA. Third, we parameterize more complex, heterogeneous, or less well understood pathways into simpler transformations, e.g. as is the case for N2O production by ammonia-oxidation, which follows an empirical "yield" based on the nitrification rate.

We now state these assumptions more clearly, but also evaluate them more critically in Section 2.2 (Model tracers and processes, lines 124 - 148) and in particular in a new paragraph in Section 6 (Discussion and Conclusions, lines 562 - 568). We also note that, by construction, the model is designed to be periodically revised and updated as evidence for new processes and new observational constraints become available. It is our intention to do so in future releases.

It was also somewhat unclear which measurements were used to constrain the model. It seems like some of the model solutions were constrained by rate measurements and others were not; this should be clarified in the text.

We now clarify (see below) that all model optimizations were constrained by both tracer and rate measurements.

Additionally, eqns. (5) - (10) in Section 3.1 seem to follow directly from the equations in the previous section. The authors could consider combining these two sections for clarity.

Indeed, equations are presented in a sequential way. However, we prefer to separate the main model equations, describing heterotrophic and chemolithotrophic N cycle reactions and their implementation (Section 2.3), from a description of how they are embedded into a 1-D water column advection-diffusion model (Section 3.1), and driven by a specific representation of organic matter cycling (Section 3.2). One could imagine embedding the equations in Section 2.3 to a 3-D model of ocean circulation, and couple them to a different representation of the C cycle (organic matter production and remineralization), e.g., as part of an Earth System Model. (See McCoy et al., 2023, DOI: 10.22541/essoar.167058932.27589471/v1, for a somewhat different 3-D implementation of

Finally, the authors could spend more time discussing what is potentially lost by modeling bulk reaction rates instead of individual microbial populations. In Section 2.1, "Model Rationale," the authors begin to address this, but one or two sentences about the trade-offs between the two approaches would be helpful to readers not intimately familiar with either type of modeling.

This is another excellent suggestion. We expanded the discussion of the trade offs between the two approaches in Section 6 (Discussion and Conclusions, lines 575 - 581, see below).

**Specific Comments**

the model.)

Line 33: Given that bacterial ammonia oxidizers oxidize ammonia, not ammonium, it might be more accurate to use the term "ammonia oxidation" instead of "ammonium oxidation." It remains unclear whether ammonia oxidizing archaea can oxidize ammonium as well as ammonia.

Thank you for the suggestion. We have now corrected "ammonium oxidation" to "ammonia oxidation" throughout the text. We note that we only model  $NH_4^-$  as the dominant dissolved form, and thus, in our "bulk" approach, we do not distinguish it from  $NH_3$ . We also clarify this point in the Methods section.

[Line 27] Added: ...and the main N cycle reactions consist of uptake and assimilatory reduction of  $NO_3^-$  to  $NH_4^+$  (here used interchangeably with ammonia,  $NH_3$ ), and the oxidation of  $NH_4^+$  back to  $NO_3^-$ ...

[Line 79] Added: The NitrOMZ model is based on the current understanding of the N cycle in OMZs (Lam et al., 2011; Kuypers et al., 2018) as mediated by 6 major species:  $N_2$ ,  $NO_3^-$ ,  $NO_2^-$ ,  $N_2O$ ,  $NH_4^+$ , and organic nitrogen (OrgN) in either dissolved or particulate form. We only explicitly model  $NH_4^+$  (the dominant dissolved form) and thus do not distinguish it from  $NH_3$ .

Line 35: Note that this increase in yield is often attributed to a shift from  $N_2O$  production as a byproduct of hydroxylamine oxidation to nitrifier-denitrification (Wrage et al. 2001; Stein and Yung 2003).

[Lines 36 - 37] Added: "The number of  $N_2O$  molecules produced per **NH**3 oxidized, i.e. the yield of this reaction, increases as  $O_2$  declines (Goreau et al., 1980; Nevison et al., 2003) likely caused by a shift from  $N_2O$  production as a byproduct of hydroxylamine oxidation to nitrifier-denitrification (Wrage et al., 2001; Stein and Yung, 2003)."

Lines 88-89: The authors state the advantages of the "system view" approach. But what kinds of information are lost — what are the disadvantages — of modeling chemical reactions instead of microbial populations?

We added a discussion of this issue in Section 6 (Discussion and Conclusion).

[Lines 575 - 581] Added: "Indeed, previous modeling studies have pointed out the value of explicitly resolving the biomass of microbial populations (Penn et al. 2016; Zakem et al. 2020). This, in turn, enables a more direct comparison of model results with molecular observations (Louca et al. 2016), and could favor the emergence of complex feedbacks between microbes and their substrates driven by resource competition and oceanic circulation (Penn et al. 2019). However, explicitly simulating microbial biomass requires a number of additional parameters that remain poorly constrained and adds computational burden that may not always improve the realism of biogeochemical simulations (Galbraith et al. 2015). Our model provides a valuable framework to continue exploring these ideas in both idealized and realistic settings (McCoy et al. 2022)."

Line 91: An emerging view of  $N_2O$  cycling indicates that nitrate reduction to  $N_2O$  with no exchange of nitrite with an external pool is a major source of  $N_2O$  in and around OMZs (Ji et

al. 2018; Casciotti et al. 2018). This important exception to the modularity of the N cycle should be noted.

We noted this (and the other missing processes i.e. DNRA, hybrid  $N_2O$  production) in the final paragraph of section 2.2 (Model tracers and processes).

[Lines 124 - 127] Added: "There are several notable processes that are not represented in the current model formulation, but could be introduced in future releases. Some of these processes (e.g. dissimilatory  $NO_2^-$  reduction to  $NH_4^-$ , DNRA) are not thought to be quantitatively relevant in oceanic oxygen minimum zones. Others, while relevant, require further measurements to constrain their significance and response to environmental variability."

[Lines 128 - 133] Added: "Production of  $N_2O$  via  $NH_4^-$  oxidation in NitrOMZ is represented as a single  $O_2$ -dependent function designed to model the transition in bacterial metabolisms from predominantly hydroxylamine oxidation to nitrifier-denitrification at low O2 (Wrage et al., 2001; Stein and Yung, 2003; Nevison et al., 2003). However, growing evidence suggests ammonia oxidizing archaea (AOA, which greatly outnumber their bacterial counterpart) can also produce  $N_2O$  via a hybrid mechanism (Santoro et al., 2011; Loscher et al., 2012). Production of  $N_2O$  via AOA appears to be similarly enhanced at low  $O_2$  (Trimmer et al., 2016; Santoro et al., 2021), although evidence from Stieglmeier et al. (2014) argues otherwise."

[Lines 134 - 137] Added: "DNRA, which can be dominant in anoxic sediment, has been sporadically observed in the water column of oxygen deficient zones, where it may provide an additional source of  $NH_4^-$  to anammox bacteria (Lam et al., 2009; Lam and Kuypers, 2011; Kraft et al., 2011; Jensen et al., 2011). However, DNRA is occasionally undetectable in OMZ waters (Kalvelage et al., 2013; De Brabandere et al., 2014), and its importance to OMZ N cycling is still debated (Long et al., 2021a)."

[Lines 138 - 143] Added: "Recent tracer incubation studies show substantial and often dominant formation of  $N_2O$  from  $NO_3^-$  rather than  $NO_2^-$  (Ji et al., 2018b; Frey et al., 2020). This suggests that denitrifying bacteria capable of direct production of  $N_2O$ from  $NO_3^-$  reduction (as  $NO_2^-$  reduction proceeds entirely within the cell) could be a major source of  $N_2O$ . This idea, which contrasts with the model assumption of a fully ``modular''' N cycle, is further supported by isotopic evidence (Casciotti et al., 2018). Observations needed to constrain the proportion of  $N_2O$  from  $NO_3^-$  and  $NO_2^-$  and its environmental sensitivity remain however limited (Ji et al., 2018b; Frey et al., 2020)."

[Lines 144 - 148] Added: "Finally, the model could easily accommodate missing processes that couple the N cycle with other elemental cycles, in particular carbon

and sulfur. These include formation of organic matter by chemolithotrophy, changes in inorganic carbon chemistry (e.g., pH) by anaerobic reactions (Cinay et al.), and additional metabolic pathways such as anaerobic oxidation of sulfide with NO3- (Callbeck et al., 2021) and anaerobic oxidation of methane with NO2- (Thamdrup et al., 2019) (both chemolithotrophic denitrification reactions)."

We also added new considerations on the assumption of a "modular" N cycle in Section 6 (discussion and Conclusion), specifically:

[Lines 563 - 568] Added: "The model is built around two major simplifications: the ``modularity" of the N cycle, and the representation of microbial metabolisms as ``bulk" chemical reactions that avoid explicitly tracking diverse microbial populations. Both are approximate views of the N cycle. For example, recent evidence suggests that microorganisms with the ability to carry out intracellular reduction of  $NO_3^-$  to  $NO_2^-$  and  $NO_2^-$  to  $N_2O$  may dominate production of  $N_2O$  in oxygen deficient waters (Ji et al., 2018; Frey et al., 2020), although the sensitivity of this process to environmental factors is still being uncovered."

Line 109: Explain that pathway 2b implicitly represents both  $N_2O$  production as a byproduct of hydroxylamine oxidation (Hooper and Terry 1979) and nitrifier-denitrification (Wrage et al. 2001; Stein and Yung 2003).

[Lines 112 - 113] Added: Chemolithotrophic processes consist of aerobic oxidation of  $NH_4^-$  to both  $NO_2^-$  ( $R^{no2}_{ao}$ , pathway 2a) and  $N_2O$  ( $R^{n2o}_{ao}$ , pathway 2b via both hydroxylamine oxidation and nitrifier-denitrification),...

Line 128: Explain that  $K_H$  is a rate constant first-order to POC.

[Line 157] Added: "...;  $k_H$  the specific **first-order** reaction rate  $(s^{-1});...$ "

Line 136/Eqn. (2): It would be helpful to explain why this is better than using a second-order rate constant and making the reaction second order to [X] and [Y].

We do not necessarily think our approach is better than second or higher order dependencies; it is just a simplifying approximation that incorporates a Michaelis-Menten saturation on substrate – a generally well-accepted assumption. In fact, other approaches have been applied. E.g. Paulot et al. 2020 (JAMES) adopt a higher order equation (exponent of 3) for the  $NH_4^-$  dependence of ammonia oxidation. We are not aware of studies that explicitly test the sensitivity to this type of assumption, although it would make for a worthwhile analysis.

Note that we added a more general discussion of these issues in the Conclusions (lines 569 - 574), in response to a comment by Reviewer 2.

Line 145: (are first order to the concentration of organic matter)

[Line 174] Replaced: "...proceed at a rate proportional..." with "...are first-order..."

Line 156: Also cite Frey et al. (2020).

[Line 184] Added: Frey et al. (2020) citation.

Line 159: Many of these "anaerobic" reactions are shown in this manuscript to occur at non-zero oxygen concentrations. Would it then be more accurate to refer to these reactions as "suboxic" instead of "anaerobic"?

The use of the term 'anaerobic' is chosen to facilitate the separation between aerobic reactions that require  $O_2$  (with M-M dependency on  $O_2$ ) and anaerobic reactions that do not require it, and are generally shown to be inhibited by it (exponential limitation at increasing  $O_2$ ).

Line 174: Explain that kv is different from molecular diffusion, which is much slower.

[Lines 203 - 204] Added: "...and  $K_v$  is the vertical turbulent diffusion coefficient ( $m^2 s^{-1}$ , distinct from molecular diffusion which is much smaller),..."

Lines 185-186: What is the residence time of water in the ETSP, and how do the chosen relaxation timescales compare to this residence time?

The calculated residence time of  $N_2O$  in the oxycline was 5.5 - 7 years from Ji et al. (2015), while Johnson et al. (2014) suggest a more general 1-10 years for water in the ETSP, although the actual value depends on the depths considered, i.e., generally increasing in deeper layers.

[Lines 215 - 216] Added: "...i.e., on the order of 3 years **and in agreement with recent** estimates of the residence time of water within the Eastern Tropical South Pacific (ETSP) (Johnston et al., 2014; Ji et al., 2015).

Line 217/Section 3.3: Consider merging these two sections to make the equations easier to follow.

As discussed above, we prefer to keep these three sections separate, as they discuss different aspects of model implementation, and may be easy to read through.

**Line 279: Is the ability of CMA-ES to find a global optimum relevant for a problem that has multiple optimal solutions?**

Good question. It is reassuring that the algorithm can approximate global optima when they exist. In practice, we do not know if our specific problem admits a unique global optimum; our tests show that the CMA-ES converges to distinct solutions for the same model configuration, within the prescribed tolerance limits, which we take as an indication that multiple optimal solutions (likely local rather than global optima) are acceptable for all practical purposes.

**[Line 311] Added: "(i.e., able to approximate the global optimum when it is known to exist)"**

**Line 287: Why choose a constant upwelling velocity but a variable vertical diffusion?**

This is a simplifying assumption. Both upwelling velocity (w) and vertical turbulent diffusion (Kv) are likely to vary with depth. We found that, in practice, the ratio between the two guantities, z\*=K,/w, provides the vertical scale for the variation of tracers under advection-diffusion balance (e.g. a passive tracer). A decrease in this vertical scale in the model helps maintain sharp oxycline in the upper boundary of the OMZ. We achieve this by varying the diffusivity rather than the vertical upwelling mostly for simplicity, because variations in vertical velocity imply divergence of water and thus, technically would need to be balanced by lateral transport (or, put it another way, would require sources and sinks of tracers in the water column). There is evidence that Kv does increase with depth in tropical OMZs, e.g., as a consequence of weaker stratification, and is stronger in the upper thermocline (e.g. Fischer et al., 2013, Biogeosciences, Brandt et al., 2015, Biogeosciences, for the Tropical North Atlantic OMZ). However these patterns are also regionally variable, and may depend on the horizontal circulation (which can drive vertical shear). We also note that the 1-D framework conflates vertical and horizontal processes, in particular transport along sloping isopycnals. As such, we take it as a simplifying assumption that provides a reasonable representation of water column transport leading to realistic distributions of tracers, while not necessarily reflecting the underlying (three-dimensional) physics accurately.

[Lines 319 - 322] Added: "This is a simplifying assumption that allows us to control the vertical scale for advective-diffusive transport (given by the ratio between vertical diffusivity and upwelling velocity,  $K_v/w_{up}$ ), without requiring vertical divergence terms in the conservation equation for tracers associated with variable vertical velocities."

Lines 288-289: Explain why targeting the core of the ODZ justifies turning off far-field restoring.

This is another simplifying assumption that reflects the idea of sluggish horizontal circulation in OMZs (e.g., as reviewed by Karstensen et al., 2008, Prog. Ocean).

[Lines 322 - 323] Added: "Since this simulation targets the core of the OMZ, generally characterized by sluggish horizontal circulation (Karstensen et al., 2008), we turn off far-field tracer restoring."

Line 293: Is "anoxic" the right word here if we're talking about water that still has dissolved oxygen (albeit in very small concentrations)? I would suggest "oxygen deficient" as an alternative.

[Line 328] Replaced: "anoxic" with "oxygen deficient".

Lines 294-295: Is the mixed layer depth also specified in the model? Because I would expect this to affect the oxygen profile as well.

In the model, we assume that the top grid cell is just below the mixed layer, for which we specify a depth (here, 30m below the surface, a typical mixed layer depth of tropical waters). However, note that changing this depth would not directly impact the results (although it could require updating top boundary conditions for tracers and particulate organic carbon). We updated the text to clarify this.

[Lines 248 - 249] Added: "For the purpose of testing and illustration, we implement NitrOMZ in a 1D representation of the water column **below the mixed layer**, following previous work (Babbin et al., 2015)."

[Line 256] Added: "As in Babbin et al. (2015), **NitrOMZ does not represent primary production in the surface layer, and is instead** forced at the uppermost boundary by a flux of sinking POC..."

Lines 308-309: Doesn't this indicate that far-field restoring should be turned on?

It does, but, as discussed in the Methods, this would add more complications, i.e., a restoring term to represent lateral transport that varies with depth. Rather than introducing poorly-constrained lateral transport terms, we decided to keep a simple water column transport parameterization that still provides reasonable profiles of  $NO_3^{-1}$  and  $PO_4^{3-1}$ , while assigning higher weight in the optimization to the (more) locally-generated N\*.

Lines 309-310: How does weighting N\* instead of nitrate and phosphate individually work around the discrepancies between modeled and observed nitrate and phosphate?

In a sense, the information provided by  $NO_3^-$  and  $PO_4^{3-}$  is the same as that provided by N\* (plus at least one of the two nutrients). However, the N\* signal is mostly generated within the OMZ in our simulations, and directly reflects local denitrification, and thus is more important by itself to constrain the extent of denitrification. Given the evidence for some level of lateral transport in the nutrient profiles, which we decided not to include for simplicity, we opted to reduce the weight assigned to nutrients (while still recovering realistic profiles) and increasing the weight assigned to N\*, as a measure of local denitrification.

For completeness, we added a new figure to the SI (Fig. C3) that shows the model-observation discrepancies in  $NO_3^-$  and  $PO_4^{3-}$ , and referenced it in the text at line 371 and in the caption of Fig. 5.

**Lines 311-312: What about centering the vertical weight curve around the oxic-anoxic interface, where we would expect the greatest overlap between aerobic and suboxic processes?**

That's an interesting suggestion, which we haven't tested. We weighted strongly the OMZ core, because that provides more control on the OMZ thickness and the overall strength of anaerobic processes and resulting tracer peaks (N\*,  $NO_2^{-}$ ). We found that, to first order, the strength and thickness of the oxygen deficient core (for a given POC flux) control the strength of these processes and the resulting tracer signatures. Weighting the boundaries of the OMZ more strongly could allow a better simulation of  $N_2O$  maxima, which is indeed a desirable feature of the solution. However, that could come at the cost of worse  $NO_2^{-}$  and  $N^*$ , which are first order tracers for anaerobic transformations.

[Line 346] Added: "This vertical weight is shaped as a Gaussian curve centered at the core of the observed OMZ, where the bulk of anaerobic transformations targeted by our model occurs, so that values within the core of the OMZ are weighted up to twice as much as values outside the OMZ."

**Lines 324-325: Is it a problem that these parameters are converging at the boundary?**

In general, yes. This could signal that either (1) the ranges are too narrowly specified; (2) the optimization cannot recover realistic parameter values (i.e., within the range), including because of errors/uncertainties in the data and cost function; or (3) the model is not able to reproduce observations with realistic parameter values. We cannot exclude any of these possibilities (that would perhaps require specific tests with synthetic data). We also note that often reductions in the cost function that eventually push these parameters to the extremes of their prior ranges are sometimes relatively small (i.e., close to the tolerance level).

We consider solutions with parameters close to the limit of their range as reasonable given our initial assumptions (i.e., model formulation and priors), but this result does indicate that additional investigations are needed. We argue in the discussion that observational studies should target the values of these parameters to better constrain priors. We also recognize that model parameters are not necessarily well-defined "physical constants", and in fact encapsulate the response of a complex system to environmental conditions. Thus the broad applicability of specific constant values may not always be accurate.

**Line 316: Are there the biogeochemical implications for these trade-offs?**

This is an excellent question. By construction, these trade-offs generally produce similar tracer distributions and do not affect the overall cost function. However, some of them may lead to trade-offs in tracer or rate profiles, meaning that improving one relative to observations may worsen another.

**Lines 334-335: This seems somewhat circular. They should show similar profiles, right? If they're being fit to profiles of data that have the same shape?**

Indeed, to some extent we expect similar profiles as a result of the optimization. However, given that the cost function reflects a global fit to multiple variables at multiple depths, the same cost function could be obtained with different features of the solutions, e.g., by improving the fit to one variable while worsening the fit to another. To some extent, this happens for few optimizations (as shown by gray lines in Fig. 5 that more markedly diverge from the other solutions).

[Lines 369 - 371] Added: "Considering the variability in the optimal parameter sets, and the complexity of the cost function, which depends on observations for multiple variables at different depths, the resulting N-cycle profiles show similar features across all optimal solutions (Fig. 5, top panels; see also Fig.C3 for macronutrient profiles)."

Lines 338-339: Could some of these discrepancies be attributed to the fact that the rate data are potential rates, especially in the cases where the model produces smaller rates than the measured ones?

This is a good point; we added a clarification to the text.

**[Lines 375 - 376] Added: "Lower rates than observed may also reflect the fact that incubation experiments provide potential rates, rather than in situ rates."**

Line 339: "Indirect" could potentially be confusing here since rate measurements with 15N tracer are often referred to as "direct rate measurements."

Removed. Indeed, our intention was to convey the complex nature of these measurements, which may not always reflect in situ rates. (Now addressed in the added sentence.)

**Lines 358-359: It seems as though this window allows for N2O production at higher levels of $O_2$ than 3 $\mu$ M, as suggested by Dalsgaard et al. (2014) - this is an interesting result.**

Thank you for the observation. Indeed, the  $O_2$  "window" resulting from the sequential inhibition by  $O_2$  of  $N_2O$  and  $N_2$  production by denitrification is consistent with measurements by Dalsgaard et al. 2014, although we find that the width of this window is several times larger than discussed in that paper. However, we also note that Ji et al., 2018, found inhibition thresholds several times higher than Dalsgaard et al., 2014. (As did other studies of anaerobic N-cycle processes, e.g., Kalvelage et al. 2010, for anammox.)

[Lines 395 - 399] Added: "This  $O_2$ -driven decoupling of anaerobic reactions is consistent with the observed sequential inhibition of  $N_2O$  and N2 production in incubation experiments (Dalsgaard et al., 2014), although we find  $O_2$  inhibition thresholds that are somewhat higher than suggested by those experimental studies. Conversely, other studies have suggested much higher  $O_2$ -inhibition thresholds for anaerobic processes, on the order of several mmol/m3 (Kalvelage et al., 2014; Ji et al., 2018)."

Line 378-379: What does it mean that most of the optimizations were not able to reproduce the observed  $N_2O$  profile, and that the one that does appears to be an edge case with respect to many of the transformation rates in Figure 5 and the proportions of total N loss in Figure 7? Is this evidence for processes missing in the model?

This is another interesting observation. While all optimizations reproduce the approximate shape of N2O across the OMZ, only a subset reproduce the relatively large magnitude of the secondary N2O peak at the lower oxycline. This subset belongs to a "cluster" of optimizations characterized, among other features, by (1) higher O2 inhibition thresholds than other clusters (between 1.0 and 2.0 mmol/m3 for NO2- reduction, and between 0.5 and 1.0 mmol/m3 for N2O reduction); (2) a wider O2 "window" where net N2O production is favored (between 0.5-1.0 mmol/m3 width). These features enable (1) a somewhat deeper reach of anaerobic reactions in the lower layers of the oxygen deficient zone, which leads to greater rates of both NO2- and N2O reduction, and, critically, a (2) larger excess of N2O production that favors deep N2O accumulation (Fig. 6).

It's interesting to note that not all optimizations, even though they share the same cost function, reproduce the same magnitude for this deep  $N_2O$  maximum and other features. We attribute these discrepancies to other characteristics of the cost function being more

heavily weighted than the deep  $N_2O$  peak location and magnitude. This specific example highlights that in non-linear problems with complex, multi-variable cost functions, subsets of optimal parameters with similar features can emerge, resulting in more desirable solutions. To identify optimal parameter sets more easily, one could revise the cost function to place more emphasis on these features. That's why we increased the weight given to  $N_2O$  relative to other tracers. We have now made changes to the text to clarify these points.

[Lines 421 - 426] Added: "Among tracers,  $N_2O$  profiles show significant variability between optimizations. While all optimizations generate two peaks in  $N_2O$  surrounding the oxygen deficient core, only a subset is able to reproduce the observed magnitude of the secondary peak at the lower oxycline (roughly 500 m depth, see Fig. 5). This subset forms a "cluster" of optimizations that share common features that facilitate the formation of a realistic deep  $N_2O$  peak, including higher  $O_2$  inhibition thresholds (between 1.0 and 2.0 mmol/m3 for  $NO_2$ - reduction, and between 0.5 and 1.0 mmol/m3 for  $N_2O$ reduction), and a wider  $O_2$  window where net  $N_2O$  production is favored (between 0.5 and 1.0 mmol/m3 width)."

Line 385-386: Is this indicative of an ammonia-oxidizing archaea-dominated regime, as opposed to ammonia-oxidizing bacteria?

This is also an intriguing observation, as AOA are generally characterized by slower growth rates but higher substrate affinities than AOB. However, since our model takes a "bulk" approach to only resolve seawater chemistry, we do not feel confident in interpreting this aspect of the optimization in relation to microbial ecology. Nonetheless, this observation is in line with the idea that AOA dominate in this region.

Lines 384-395: Again, these trade-offs are discussed primarily in terms of the mechanics of the model, but could they also represent biogeochemical trade-offs? For example, competition between anammox bacteria and nitrite reducing denitrifiers for nitrite?

We do explain the trade-offs in terms of model behavior. However, they also represent biogeochemical trade-offs, that is, they reflect simultaneous changes in N transformation rates that result in different degrees of tracer accumulation in the water column. These changes are in turn caused by differences in model parameters that encapsulate "bulk" characteristics of microbial populations, such as affinities to substrates, inhibition by O2, and maximum metabolic rates. In a sense, while we do not model explicit microbial competition, we do represent trade-offs that would lead to specific metabolisms to dominate depending on environmental conditions and substrates.

Line 394: Specify that these half saturation constants are for anammox — it could be confusing since you mention a different set of half saturation constants above.

[Line 442] Added: "...likely because of higher anammox half saturation constants..."

Lines 421-422: This relates to similar results in Frey et al. (2020).

[Lines 477 - 478] Added: "...consistent with observations of these processes in the Peruvian oxygen deficient zone (Frey et al., 2020)."

Line 425/Section 5.4: I know it's outside of the scope of the paper to quantify climate impacts, but is it possible to tie the results in this section to predicted climate-related impacts in the ocean, such as deoxygenation, warming, and increased stratification?

Indeed, sensitivity to environmental conditions neatly ties in to the response to climate variability and change.

[Lines 484 - 485] Added: "Critically, these parameters are likely to vary over time under the effects of natural climate variability (e.g., Deutsch et al., 2011), and anthropogenic climate change Bopp et al., (2013)."

[Lines 487 - 488] Added: "Perturbations that replenish  $O_2$  above the thresholds for anoxic processes --- such as those predicted under climate warming scenarios (Busecke et al., 2022) --- have thus cascading impacts on anaerobic N cycle intermediates ..."

Lines 426-427: What about mixed layer depth?

[Line 483] Added: "The main features of the OMZ simulated by the model are strongly dependent on environmental parameters such as upwelling and mixing, organic matter fluxes, and the model boundary conditions, **including mixed layer depth and O**2 **concentrations.**"

Lines 429-430: ... and so too do the thresholds, themselves.

Indeed, the interaction between environmental conditions and  $O_2$  thresholds (discussed in the "Sensitivities to model parameters" section just above) will drive the response of the low- $O_2$  N cycle.

Line 440: ...and yet a small *increase* in the magnitude of the near-surface  $N_2O$  maximum when  $K_v$  is increased - why?

Another interesting observation. Two reasons are likely behind this increase in the upper max in  $N_2O$  as Kv decreases. As the OMZ expands at lower Kv, it also shifts upward, where

it intercepts higher fluxes of sinking POC. Thus, net N2O production happens at a higher rate. Furthermore, lower Kv reduces the transport of N2O away from this peak, favoring N2O accumulation.

[Lines 500 - 504] Added: "The interplay between the position of the oxygen deficient layer, sinking particle fluxes, and transport processes further modulates the response of tracer profiles. For example, as anoxic waters expand upwards following a reduction in Kv, they intercept a higher concentration of sinking organic matter, which in turn fuels higher remineralization rates. Together with reduction in diffusive fluxes, this likely favors the strengthening of the upper N2O maximum at low Kv observed in Fig. 11."

Line 452/Section 6: The discussion feels rather short relative to the richness of the results presented above. I would have liked to see how the results (particularly regarding trade-offs and sensitivities to model parameters and environmental variables) relate to recent experimental results.

Indeed, we tried to strike a balance between a more complete and concise Discussion section. That said, we expanded the comparison of some of our results to observational studies, both in the Results and Discussion sections.

Line 460: "Capturing the correct underlying dynamics" seems like an overly strong assertion - the underlying dynamics of the model are able to capture observed tracer distributions, but this does not rule out the potential for other formulations of the N cycle to capture tracer distributions equally well.

We agree – the model is an approximation to a complex system, and structurally different formulations could lead to comparable results. That said, we are encouraged by the model's agreement not just to tracer profiles, but also – to some degree – observed transformation rates, which lends credibility to the model results. Additional observational constraints (e.g., isotopes) may be able to strengthen this confidence in the future.

[Line 523] Added: "However, by matching observed reaction rates to a reasonable degree, the model approximates the complex dynamics of the system in a way that allows it to reproduce tracer distributions."

Line 461: The model contains both aerobic and anaerobic processes. Change this to "the N cycle in and around anaerobic environments."

[Lines 524 - 525] Added: "...an effective way to constrain the model representation of the **N** cycle in and around  $O_2$ -deficient environments."

Lines 463-464: So some of the model solutions are constrained by rate measurements and some are not?

Sorry for the confusion, we didn't mean to imply that rates weren't included in the ETSP optimization procedure. We clarified this in the text, and pointed to Table B4 where these weights are found.

[Line 341] Added: "First, **all rates are weighted equally, whereas** different weights are assigned to each tracer, ..."

[Line 528] Added: "Even when rate measurements are used to constrain the model, **as done here,** an ensemble of equally good solutions is thus possible."

Lines 495-505: Add to this list additional N cycle pathways, such as DNRA, hybrid  $N_2O$  production, and  $N_2O$  production directly from nitrate.

[Lines 582 - 584] Added: "These include: (1) Additional known N cycle pathways, and their sensitivity to environmental variability, such as DNRA (Lam et al., 2009), hybrid  $N_2O$  production from AOA (StiegImeier et al., 2014), and direct  $NO_3$ - reduction to  $N_2O$  (Ji et al., 2018; Frey et al., 2020); (2) Alternative oxidation pathways...."

**Figures and tables**

Table 1: Explain why some values are N/A. Are these parameters that are optimized for in the model?

Updated Caption of Table 1: "Table 1: Summary of the main NitrOMZ parameters, with **any reported** values from the literature (**not available or 'N/A' otherwise**)."

Figure 7: It would help to show a panel with the total rate of N loss.

We included a profile of total N loss, and slightly expanded the discussion of this Figure.

[Lines 414 - 415] Added: "In the anoxic core, relief from  $O_2$  inhibition allows  $NO_2^$ reduction to outcompete anammox for  $NO_2^-$  and contribute up to 60% of the total N loss, with anammox making up the remaining 40% (also see Fig. 5)."

Figure 8: Clarify what "-50m" means.

[Line 449] Added: "These include concentrations of  $NH_4^-$  and  $NO_2^-$  at 50 m depth,..."

Updated Caption of Fig. 8: "Concentrations of  $NH_4^-$  and  $NO_2^-$  at 50 m depth (-50 m) are used as proxies of near-surface values."

**Figure 9: Why would increasing kden2 actually increase N\*?**

By increasing  $k_{den2}$ , there is slightly more POC consumption in the OMZ by  $NO_2^-$  reduction. This reduces the flux of POC to depth, and as a result the OMZ thins slightly, limiting the amount of  $NO_3^-$  that is reduced in the water column (i.e. leading to higher N\*, i.e., less negative).

[Lines 468 - 473] Added: "Notably, by increasing  $k_{den1}$  (top panels in Fig. 9) or  $k_{den2}$ (middle panels) from Optsel values, the vertical extent of anoxic waters is reduced as a result of increased POC consumption via denitrification (not shown). This enhances aerobic remineralization and nitrification below the OMZ, providing a source of NO3- that partly offsets the OMZ losses seen via  $k_{den1}$  enhancement. This may indicate a potential negative feedback: if denitrification is locally enhanced (i.e. via increased competition for POC by denitrifying heterotrophs), a resulting reduction in the vertical extent of the OMZ would inhibit further N loss."

Figure 9: There's an interesting asymmetry in the responses of  $N_2O$  to changing  $k_{den2}$  and  $k_{den3}$ . Why is this?

It's likely due to the relatively high half-saturation value for N2O reduction (0.16 mmol N2O m-3) in the Optsel parameter set. The other denitrification steps (1.0 mmol NO3- m-3 for NO3- reduction, 0.01 mmol NO2- m-3 for NO2- reduction) are easily exceeded in the water column, so the rate magnitude for those steps responds linearly to changes in the maximum rate (hence the symmetric behavior). This is not the case for N2O reduction (e.g. the water column N2O rarely exceeds the half-saturation constant), so the response is amplified at higher N2O.

Figure 12: Clarify in the caption that more negative numbers correspond to an increasing POC flux.

Updated Caption of Figure 12: "In the top panels, more negative values of  $\phi^{top}_{poc}$  correspond to an increasing sinking POC flux."

**Technical corrections** Line 219: typo: "discretized"

Fixed

Line 298: Typo: "algorithm"

**Fixed**

Line 300: Typo: "optimize"

Fixed

Line 305: Grammar: "from always converging"

Fixed

**Reviewer #2**

Apologies for the late review.

Bianchi et al. present a highly coherent and well written account of a newly developed motel formulation of nitrogen cycling in oxygen deficient zones of the ocean. Their formulation consists of functional reactions within the nitrogen cycle, parameterized as a function of the available carbon pool (POC), Michaelis Menten saturation sensitivity to substrate pools, and sensitivity to molecular oxygen. Optimizations of a 1D simulation reveals high coherence among runs, wherein salient biogeochemical features of oxygen deficient waters are adroitly represented, including nitrite and nitrous oxide.

The model parameterization is an important step forward in aiding the representation of nitrogen in the global ocean, and may enable simulations of the response of different environmental forcings to the production of  $N_2O$ , an important greenhouse gas.

I have no criticisms of the work per se, but do have questions that the authors may want to clarify in their work:

(a) What was the stoichiometric representation of nitrification and anammox? I may have missed it, but did not see it in the text.

The model calculates the stoichiometry assuming the oceanic ratios from Anderson and Sarmiento (1994), with the option to alter these ratios in the model configuration, and follows the procedures outlined in Paulmier et al. (2009). I made a note of this in Appendix A4.

[Lines 617 - 618] Added: "The stoichiometry of heterotrophic redox reactions is based on an electron balance and follows the procedure outlined in Paulmier et al. (2009), under the assumption that the composition of organic matter (POC) follows the average oceanic ratios from Anderson and Sarmiento (1994):  $C_{106}H_{175}O_{42}N_{16}P$ ."

[Lines 624 - 625] Added: "For nitrification, the oxygen to nitrogen ratios for  $NH_4^-$  and  $NO_2^-$  oxidation ( $R_{ao}$  and  $R_{no}$ , respectively) are based on the stoichiometry of the relevant redox reactions:"

$$Q^{O:N}_{ao} = (3/2)$$

 $Q^{O:N}_{no} = (1/2)$

[Line 629] Added: "Finally, for anammox,  $NH_4^-$  and  $NO_2^-$  are combined in 1:1 ratios to produce N2."

(b) How do the authors justify their respiratory quotients for respiration and denitrification, which are arguably outside empirical limits (those presented in the supplements)?

Our respiration quotients are consistent with those discussed in Paulmier et al. (2009), and are based on an electron balance for the oxidation of organic matter with different oxidants. Assuming that the bulk organic matter composition follows the oceanic ratios from Anderson and Sarmiento (1994) yields an aerobic respiration quotient ( $r_{-O2:C}$ ) of 1.11 (excluding nitrification, which is treated as a separate reaction by the model), which is within range of estimates from meridional transects across the Pacific (Moreno et al., 2020) and Atlantic (Moreno et al., 2022) basins.

**[Line 623] Added: "This yields a respiration quotient of $r_{.o2:c}$ of 1.11, which is within range of direct chemical measurements of $r_{.o2:c}$ from Moreno et al. (2020, 2022)."**

**(c) How sensitive are the simulations to said respiratory quotients?**

We did not test this within the manuscript. Increasing the respiration quotient ( $r_{-O2:C}$ ) should increase  $O_2$  consumption and expand the OMZ, which would lead to more influence from anaerobic processes. Here we use a stoichiometry based on the oceanic ratios from Anderson and Sarmiento (1994), which are commonly applied in BGC models. However, because the model includes a separate stoichiometry module that allows the user to specify different compositions of organic matter, ideas regarding changes to respiration quotients can be easily tested by varying the composition of organic matter.

(d) One limitation of the model parameterizations that I see is that maximum rates need to be prescribed ("k" values). In reality, maximum rates will depend on the abundance of the functional group of organisms in the water column. This likely explains the high sensitivity of the simulations to prescribed k values, which, in reality, will be depth dependent (e.g, more nitrifiers directly at the base of the euphotic zone than elsewhere in the water column). While the authors allude to this, I think it merits more discussion.

This is an excellent point, which we initially considered as we developed the model, although to keep it simple we eventually opted for the simpler first-order formulation presented here.

However, we recognize that "higher-order" substrate dependencies may be more appropriate to represent variability in maximum reaction rates due to (non-resolved) microbial biomass variations. This is something that we plan to address in future versions of the model. For the time being, we also note that the current version of the model produces realistic solutions within the range provided by observations, suggesting possible compensations between model parameterizations and the resulting output, such as subtle variations in tracer concentration profiles that maintain overall realistic rates in the water column.

We now address this issue more specifically in the Section 6 (Discussion and Conclusions):

[Lines 569 - 574] Added: "Our ``bulk'" approach assumes that metabolic reaction rates are proportional to substrates following a Michaelis-Menten dependency. However, in reality, reaction rates also depend on the abundance of microorganisms present in the water column. If microorganism biomass is assumed to be proportional to substrates, then a higher-order dependency of reaction rates may be more appropriate, as adopted by some biogeochemical models (e.g., Paulot et al., 2020). A different dependence on substrates, in turn, may affect the variability of reaction rates with depth, and the model sensitivity to the values of the maximum reaction rates."

**(e) I'm not sure I understood the sentence at line 340.**

Sorry for the confusion, we meant to imply the yield of N2O from ammonia oxidation is on the order of 1% (or less).

[Lines 377 - 378] Added: "In general, the yield of  $N_2O$  from  $NH_4^-$  oxidation ( $R^{n2o}_{ao}$ ) is O(100) times less than for  $NO_2^-$  ( $R^{no2}_{ao}$ ) following Equations A8 and A9, consistent with observations."

On a minor note, the reference list ascribes all findings in OMZs to a select number of recent papers and reviews. I urge the authors to acknowledge original work where pertinent.

We strived to include both primary and more recent references in the revision.

---

## Author Response (AR2)

General Statement

In their revised manuscript, Bianchi et al. addressed all of my major concerns. This paper is an excellent contribution to the field and I recommend publication after considering the few minor points below.

Figure 1/Appendix A: Have the authors considered a model scenario in which nitrite oxidation can occur anaerobically? Some studies indicate the importance of NO2– oxidation in apparently anoxic waters in ODZs (Buchwald et al. 2015; Sun et al. 2021).

This is an intriguing comment. In the current formulation of NitrOMZ, we do not consider anaerobic nitrite oxidation, partly because of the large uncertainties that surround this putative process. Instead, we assume that nitrite oxidizers are obligately aerobic organisms. It is entirely possible that nitrite oxidation occurs with oxidants other than oxygen; however, the actual pathways and chemical species involved remain unclear. For example, previous work suggested the potential for iodate reduction coupled to nitrite oxidation, or even nitrite dismutation. Both pathways could be incorporated in the model, although we feel that at this stage observations are too limited to properly constrain them, and we leave this work to future studies. We note now the possibility for anaerobic oxidation pathways in the revised manuscript.

There is also the possibility that some nitrite oxidation rates from apparently anoxic waters in ODZs may reflect the presence of vanishing oxygen concentrations. Bristow et al., (2016) demonstrated that nitrite-oxidizing bacteria (NOB) have a higher affinity for $O_2$ compared to ammonia-oxidizing archaea (AOA). This helps explain the depth distributions of AOA vs NOB, wherein NOB are typically distributed deeper and at higher abundances into the OMZ core. Buchanan et al. (2023) recently explored this relationship further, demonstrating that NOB can thrive in OMZs due to infrequent intrusions of oxygen. Thus, NOB may be adapted to take advantage of a 'goldilocks zone' of episodic oxygenation and high nitrite supply (via nitrate reduction), since their high maximum growth rates (compared with other OMZ microorganisms) allow them to grow during $O_2$ pulses, offsetting losses during fully anoxic conditions.

In a complementary study where we embed NitrOMZ into an eddy-resolving simulation of the ETSP, we similarly find time-averaged nitrite oxidation rates throughout the OMZ as a result of eddy-driven $O_2$ injections into the OMZ, which suggests that NOB growth via alternative oxidants is a complementary, but not necessary, mechanism.

To reflect these points, we added the following text to Section 2.2 "Model tracers and processes":

**Added [line 145]: "***Other work suggests the occurrence of NO2 oxidation in apparently O2-deficient waters (Buchwald et al. 2015; Babbin2020, Sun et al. 2021). This may involve NO2 oxidation coupled to iodate reduction, or NO2 disproportionation --- two poorly characterized processes. It may also reflect the high affinity to O2 of nitrite-oxidizing bacteria (Bristow, 2016) in regions where vanishing O2 concentrations are maintained by infrequent lateral intrusions (Buchanan et al., 2023).*"

Note that we mention iodate reduction and NO2 oxidation coupled to iodate reduction, or NO2 disproportionation also in the closing paragraph of the Discussion and Conclusions, where we outline future work.

Line 80 and Appendix 4: How sensitive are the model results to the assumed stoichiometry of organic matter? The C:P of organic matter in the eastern tropical South Pacific has been hypothesized to be closer to 83:1 (Teng et al. 2014) (and if the authors disagree with Teng et al. from a modeling perspective, I'd love to hear why). Changing C:P ratio should also change N:P, since C:N remains fairly consistent (Martiny et al. 2013). I know that the assumed N:P ratio can have a significant impact on N fixation rates (Mills and Arrigo 2010), although this was a very different modeling framework. On a related note, would it affect the model results for denitrification to remineralize organic matter with a different stoichiometry from aerobic respiration? I'm thinking of the Van Mooy paper showing that the stoichiometry of remineralization via denitrification may not involve PO43- because it's mostly amino acids (Van Mooy et al. 2002).

Indeed, we acknowledge that there is potential variability in the stoichiometry of organic matter in the ocean, as the Reviewer suggests, and that the use of a fixed stoichiometry is only an approximation (although still fairly common in ocean biogeochemical models). Stoichiometric variations should not qualitatively affect our results, and certainly different organic matter stoichiometries, or even differential remineralization rates for different elements, could be incorporated in the model.

We now clarify this point in the paper both in the "Model Rationale" Section, and in the Appendix, where we added the references suggested by the Reviewer.

**Added [lines 80-81]:** *"although variable stoichiometry can easily be accommodated***"

**Added [lines 625-628]: "***For example, studies in the Eastern Tropical South Pacific suggest a C:N ratio closer to 83:1 (Teng et al. 2014). Furthermore, organic matter degradation reactions may also differentially remineralize C, N and P. For instance,*

*denitrification may preferentially involve degradation of amino-acids, and thus impact the N:P ratio of remineralization differently from aerobic respiration (Van Mooy et al. 2002).***"**

Line 343: I might be misunderstanding, but should this say "a random error of up to 20%"?

The Reviewer is correct, "*up to 20%*" is the correct way to describe our approach. We added this to **line 343.**

Line 627: Why only two electrons to reduce 2NO2- to N2O?

In the stoichiometry module, we assume:

$$e^- + NO_2^- + 2H^+ \rightarrow NO + H_2O$$

$$e^- + NO + H^+ \rightarrow 1/2\ N_2O + 1/2\ H_2O$$

Since we do not represent NO in the current version of NitrOMZ, the summation would be represented as:

$$2e^- + NO2^- + 3H^+ \rightarrow 1/2\ N_2O + 3/2\ H2O$$

Therefore, we use 2 electrons to reduce $1NO_2^-$ to $\frac{1}{2}N_2O$, which is accounted for by the sources-minus-sinks module. We added a ½ to clarify.

**Added [line 637]:** For denitrification, 2 electrons are required for each respective reduction step (NO3 to NO2 , NO2 to ½N2O, and N2O to N2)...

References
Buchwald, C., A. E. Santoro, R. H. R. Stanley, and K. L. Casciotti. 2015. Nitrogen cycling in the secondary nitrite maximum of the eastern tropical North Pacific off Costa Rica. Glob. Biogeochem. Cycles 29: 2061–2081. doi:10.1002/2015GB005187

Martiny, A. C., J. A. Vrugt, F. W. Primeau, and M. W. Lomas. 2013. Regional variation in the particulate organic carbon to nitrogen ratio in the surface ocean. Glob. Biogeochem. Cycles 27: 723–731. doi:10.1002/gbc.20061

Mills, M. M., and K. R. Arrigo. 2010. Magnitude of oceanic nitrogen fixation influenced by the nutrient uptake ratio of phytoplankton. Nat. Geosci. 3: 412–416. doi:10.1038/ngeo856

Sun, X., C. Frey, E. Garcia-Robledo, A. Jayakumar, and B. B. Ward. 2021. Microbial niche differentiation explains nitrite oxidation in marine oxygen minimum zones. ISME J. 15: 1317–1329. doi:10.1038/s41396-020-00852-3

Teng, Y.-C., F. W. Primeau, J. K. Moore, M. W. Lomas, and A. C. Martiny. 2014. Global-scale variations of the ratios of carbon to phosphorus in exported marine organic matter. Nat. Geosci. 7: 895–898. doi:10.1038/ngeo2303

Van Mooy, B. A. S., R. G. Keil, and A. H. Devol. 2002. Impact of suboxia on sinking particulate organic carbon: Enhanced carbon flux and preferential degradation of amino acids via denitrification. Geochim. Cosmochim. Acta 66: 457–465. doi:10.1016/S0016-7037(01)00787-6